# Unraveling the evolutionary origin of the complex Nuclear Receptor Element (cNRE), a *cis*-regulatory module required for preferential expression in the atrial chamber

Luana Nunes Santos[1,2,3], Ângela Maria Sousa Costa[1], Martin Nikolov[2], João E. Carvalho [4], Allysson Coelho Sampaio[3,5], Frank E. Stockdale[6], Gang Feng Wang[6,13], Hozana Andrade Castillo[1,2], Mariana Bortoletto Grizante[1], Stefanie Dudczig [7], Michelle Vasconcelos[3], Nadia Rosenthal[8,9], Patricia Regina Jusuf[7], Hieu T. Nim [10], Paulo de Oliveira [1], Tatiana Guimarães de Freitas Matos[3], William Nikovits Jr.[6], Izabella Luisa Tambones[1], Ana Carolina Migliorini Figueira [1], Michael Schubert [4], Mirana Ramialison [2,10] ✉ & José Xavier-Neto [11,12] ✉

Cardiac function requires appropriate proteins in each chamber. Atria requires slow myosin to act as reservoirs, while ventricles demand fast myosin for swift pumping. Myosins are thus under chamber-biased *cis*-regulation, with myosin gene expression imbalances leading to congenital heart dysfunction. To identify regulatory inputs leading to cardiac chamber-biased expression, we computationally and molecularly dissected the quail Slow Myosin Heavy Chain III (*SMyHC III*) promoter that drives preferential expression to the atria. We show that *SMyHC III* gene states are orchestrated by a complex Nuclear Receptor Element (cNRE) of 32 base pairs. Using transgenesis in zebrafish and mice, we demonstrate that preferential atrial expression is achieved by a combinatorial regulatory input composed of atrial activation motifs and ventricular repression motifs. Using comparative genomics, we show that the cNRE might have emerged from an endogenous viral element through infection of an ancestral host germline, revealing an evolutionary pathway to cardiac chamber-specific expression.

Vertebrate chambered hearts are efficient pumps organized according to an ancient evolutionary paradigm that divides their circulatory functions into two working modules controlling inflow and outflow working, the atria and the ventricles, respectively[1]. Most current views on cardiac chamber development agree with the principles of epigenesis, in which higher-level structures arise through sequential morphogenetic steps. The gradual development of the three-dimensional structure is accompanied by progressive restriction of cellular fates, from large, early embryonic fields that initially display broad tissue potencies (e.g., epiblast and mesoderm) to terminally differentiated cells (i.e., most heart cell types). The varied cardiac cell fates originate from a combination of mosaic and regulative developmental processes[2]. Ultimately, these developmental processes combine clues from the relative position of each group of cells inside the embryo with the information imparted by patterning and migration, which modify the initial relationships between cell progenitors and further restrict fates.

Most efforts in the last twenty years have been dedicated to transpose the blueprints of the above-mentioned cardiac ontogenetic events from the four-dimension arena into overlapping chemical (i.e., signaling) and genomic spaces, with a strong focus on gene regulatory pathways[3]. However, a clear description of these ontological and genomic events analogous to an engineering blueprint is still elusive. Moreover, it has become clear that

---

another dimension will have to be accounted for, that is evolutionary time, whose major arena is, yet again, the genome. Rather than being a uniform environment in which mutations occur sporadically, the genome is a lively space, often changed by the insertion and proliferation of mobile genetic elements, such as viruses and transposons[4]. All these roving actors have the potential to modify the genome, creating sequence combinations that may take on novel roles in gene regulation[4].

To understand the establishment and the evolution of cardiac gene regulatory networks, we have previously examined the Slow Myosin Heavy Chain III (*SMyHC III*) gene promoter and determined that it drives preferential atrial gene expression in quail and mice[5–10]. The sequence 5'-AGGACAaagAGGGGA-3' located from −801 to −787 base pairs (bp) upstream from the transcription start site of *SMyHC III* contains two hexad sequences (Hexads A and B). Hexads A and B were previously identified as a dual Vitamin D Receptor Element (VDRE) and a Retinoic Acid Receptor Responsive Element (RARE)[6–8], respectively, with a ventricular inhibitory function associated with the VDRE in the quail[6]. The *SMyHC III* promoter further contains a GATA-binding element required for activating expression in both atrial and ventricular quail cardiomyocytes[7]. However, and to our surprise, mutation of either the VDR, or the GATA site did not affect preferential atrial expression of the *SMyHC III* promoter in transgenic mice[9].

We thus sought to investigate if there are additional mechanisms for preferential atrial expression by the *SMyHC III* promoter using in silico and in vivo approaches. We found that the atrial preference exhibited by the *SMyHC III* promoter is conserved in avians, teleost fishes, and mammals, chiefly on account of a low frequency repetitive 32-bp genomic element formed by tandem reiterations of three purine-rich hexanucleotide repeats, here designated as the complex Nuclear Receptor Element (cNRE). The cNRE is a versatile regulator of selective cardiac chamber expression, switching from *SMyHC III* activator to repressor functions according to atrial or ventricular contexts, respectively. We demonstrate that the combination of three hexanucleotide repeats within the cNRE, the classic Hexads A and B containing a ventricular inhibitor[6–8] plus a novel Hexad C encompassing an atrial activator, provides an information processing platform that integrates different signals to bias gene expression towards an atrial readout. Finally, using comparative genomics, we provide evidence that the cNRE was associated with the *SMyHC III* gene in the Cretaceous, between 60 and 70 million years ago, possibly by recombination of an unknown virus into the genome of an ancestral galliform bird[11].

## Results

### The cNRE displays a tripartite structure with Hexads A, B, and C

To investigate mechanisms of atrial specificity, we performed computerized profiling of nuclear receptor binding sites in the *SMyHC III* promoter. Computerized profiling is a simulation approach to identify nuclear receptor binding sites (Hexads) using Poisson–Boltzmann's theory to calculate interaction energies between nuclear receptor proteins and DNA as an approximation of their respective binding affinities. The profiling predicted a novel nuclear receptor binding hexad, Hexad C (5'-gaaggacaaa-gagggggacaaagaGGCGGAggt-3' at −780 to −775 bp), adjacent to Hexads A and B known to act as ventricular repressor sequences[6,7] (Supplementary Fig. S1). The combination of these three Hexads sequences (A + B + C) was designated as the complex Nuclear Receptor Element (cNRE) (Fig. 1A). We next assessed whether the tripartite cNRE was able to physically bind nuclear receptors. We thus conducted fluorescence anisotropy binding assays with fluorescein-bound Hexad A, B, and C oligonucleotides and their combinations (AB, BC, AC, and ABC) using in vitro-synthesized and fast protein liquid chromatography (FPLC)-purified COUPTF-II (COUP transcription factor 2) and GR (glucocorticoid receptor) (Supplementary Fig. S2). We found that all three Hexads (A, B, and C) bound the nuclear receptor COUPTF-II with high affinities. COUPTF-II bound Hexad A with a $K_d$ of 84.1 (± 7) nM, Hexad B with a $K_d$ of 58.0 (± 1) nM, and Hexad C with a $K_d$ of 127.0 ( ± 16) nM. In comparison, the nuclear receptor GR displayed much lower affinities for the Hexads. GR thus bound Hexad A with a $K_d$ of

1028.0 (±79) nM and Hexad B with a $K_d$ of 752.4 (±19) nM, while not binding Hexad C at all. Similarly, Hexad A plus B (AB), B plus C (BC), A plus C (AC), and A plus B plus C (ABC) bound COUPTF-II with nanomolar affinities: AB with a $K_d$ of 90.7 (±9) nM, BC with a $K_d$ of 397.0 (±28) nM, AC with a $K_d$ of 367.3 (±17) nM, and ABC with a $K_d$ of 344.2 (±58) nM. GR bound the double Hexads with much lower affinity: AB with a $K_d$ of 1325.8 (±90) nM, BC with a $K_d$ of 817.2 (±189) nM, AC with a $K_d$ of 1431.5 (±357) nM, and ABC with a $K_d$ of 2,330.8 (±189) nM. We therefore concluded that the cNRE, composed of Hexads A, B, and C, is a complex nuclear receptor element that binds specific subsets of nuclear receptors with nanomolar affinities, which is consistent with its previously established roles in binding VDR, RARα, and RXRα[7,8]. We further postulated that this novel tripartite nuclear receptor binding element contains information for preferential atrial expression in vertebrate embryos.

### The cNRE is a transferable *cis*-regulatory agent needed for atrial expression

To test the requirement of the cNRE for preferential atrial activation, we performed transient expression assays in zebrafish embryos (Fig. 1B). Two reporter constructs, the quail *SMyHC III* promoter driving eGFP (*SMyHC III*::eGFP) and the quail *SMyHC III* promoter, in which the cNRE was deleted (*SMyHC IIIΔcNRE*::eGFP), were injected into Tg(*vhmc*::mCherry) embryos[12] that, within the cardiac tissue, express mCherry exclusively in the ventricle[13]. We assessed the proportion of embryos displaying eGFP expression restricted to the atrium, restricted to ventricle, or present in both chambers (Fig. 1C, D, Supplementary Data 1). We also quantified the proportion of zebrafish embryos presenting eGFP expression in noncardiac tissue (Supplementary Fig. S3). With the wild-type (WT) *SMyHC III* promoter construct, 38% of embryos (*n* = 33 of 86 embryos) displayed eGFP expression restricted to the atrium, and 52% (*n* = 45 of 86 embryos) expressed the reporter in both ventricular and atrial chambers (Fig. 1D). However, there was greater atrial-specific expression (38%) than ventricular-specific expression (9%, *n* = 8 of 86 embryos) (Fig. 1B, D). In contrast, atrial-specific expression was significantly reduced in reporter assays with the mutated quail promoter *SMyHC IIIΔcNRE*::eGFP (15%, *n* = 9 of 59 embryos, *p* = 0.0026), and, concomitantly, ventricular-specific expression was significantly increased (44%, *n* = 26 of 59 embryos, *p* = 0.0038) (Fig. 1B, D). The changes in the proportions of embryos displaying expression in both chambers for the *SMyHC III*::eGFP and the *SMyHC IIIΔcNRE*::eGFP constructs were not statistically significant (52%, *n* = 45 of 86 embryos and 41%, *n* = 24 of 59 embryos, respectively) (Fig. 1D, Supplementary Fig. S3). These results indicate that deletion of the cNRE from the *SMyHC III* promoter reduces overall atrial expression. Taken together, data from expression assays in zebrafish support the notion that the cNRE drives preferential atrial expression outside the original avian context in phylogenetically distant vertebrate species. Assays in the context of transgenic *SMyHC III*::HAP mouse lines, in which WT and mutant quail *SMyHC III* promoters drive expression of the human alkaline phosphatase (HAP)[9], corroborated this interpretation (Fig. 1E, F, Supplementary Fig. S4, Supplementary Data 1). We observed that deletion of a 72-bp region encompassing the cNRE (*SMyHC IIIΔcNRE*::HAP) markedly reduced atrial expression (Fig. 1G, H, Supplementary Fig. S4), indicating that, in addition to the previously described ventricular repressors, the cNRE contains positive regulatory elements driving expression in vertebrate atria.

### The cNRE biases expression towards a preferential atrial pattern

To test the sufficiency of the cNRE for driving preferential atrial expression, we devised a conversion assay, which aimed at testing whether the cNRE is sufficient to change a pattern of strong ventricular-specific expression towards atrial activation. To do so, we performed transient expression assays in zebrafish, with the complete 2.2 kb-long *vmhc* promoter that drives powerful ventricle-specific expression of eGFP (*vmhc*::eGFP) (Fig. 1I)[13] and with a 5' fusion of five tandem repeats of the cNRE to the *vmhc* reporter construct (*5xcNRE-vmhc*::eGFP) (Fig. 1J). At 48 h post fertilization (hpf), we observed ventricle-specific eGFP expression in most transients injected with

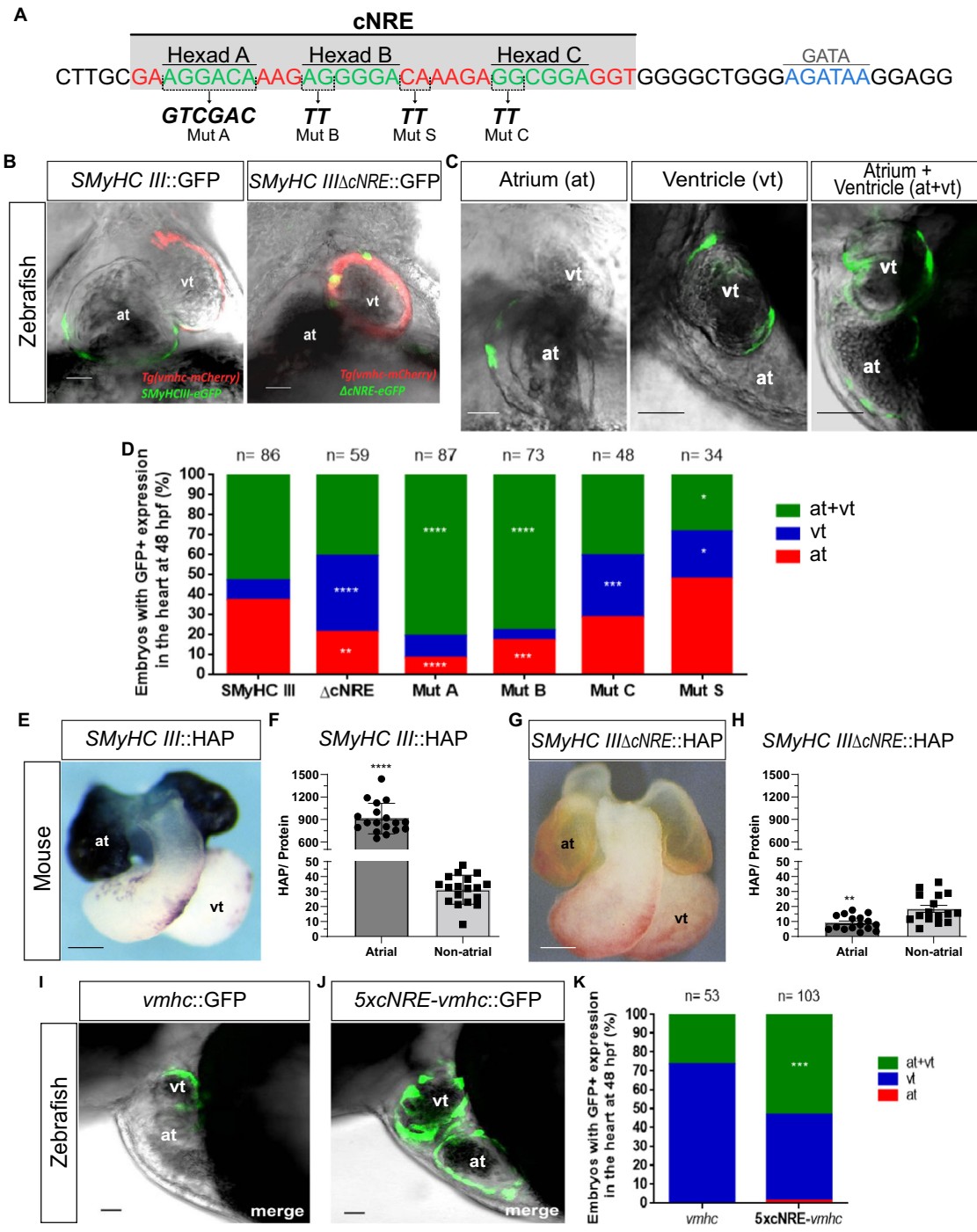

vmhc::eGFP (74%, n = 39 of 53 embryos) (Fig. 1K). There were no embryos expressing eGFP exclusively in the atrium. With the 5xcNRE-vmhc fusion construct (Fig. 1J), we observed a significant increase in the proportion of embryos showing both atrial and ventricular eGFP expression (53%, n = 55 of 103 embryos, compared to the vmhc::eGFP: 26%, n = 14 of 53 embryos), a decrease in the number of embryos expressing eGFP exclusively in the ventricle (46%, n = 47 of 103 embryos, compared to the vmhc::eGFP: 74%, n = 39 of 53 embryos), and, strikingly, a single embryo with reporter expression exclusively in the atrium (0.97%, n = 1) (Fig. 1K). These experiments demonstrated that, outside of its native context in the quail SMyHC III promoter, the cNRE is sufficient to shift cardiac expression from a state of strong ventricular specificity towards a pattern of increasing atrial expression. Taken together, our results suggest that the cNRE sways cardiac expression towards a model of preferential atrial gene expression. We hence

sought to investigate the precise contribution of binding elements within the cNRE to the activation of atrial-specific gene expression and to the repression of ventricular-specific gene expression.

### Combinatorial recruitment of Hexads A, B, and C is essential for atrial stimulation and ventricular repression

To understand the *cis*-regulatory composition of the cNRE, we assessed the contribution of Hexads A, B, and C in driving preferential atrial expression in vivo. To obtain maximal comparability with previous studies and to simultaneously probe cNRE function at an increased resolution, we utilized previously described SMyHC III promoter constructs[6] and introduced novel site-directed dinucleotide substitution mutants of individual Hexads in the SMyHC III promoter (SMyHC III::eGFP) (Figs. 1–3, Supplementary Figs. S3, S4). Reporter expression driven by the mutated cNREs was

**Fig. 1 | The cNRE drives expression in atrial cells. Mutational analysis of the *SMyHC III* promoter in zebrafish reveals a dual role in atrial activation and ventricular repression. A** Schematic representation of the *SMyHC III* promoter sequence highlighting the position of the cNRE sequence and mutated sites. **B** Confocal images in frontal views, anterior to the top, of a representative zebrafish embryo. Ventricular expression is demonstrated by overlapping eGFP expression driven by *SMyHC IIIΔcNRE* and stable mCherry fluorescence driven by the ventricular stable line. **C** Representative panel of eGFP expression patterns in cardiac chambers of zebrafish embryos in lateral views, anterior to the left, injected with *SMyHC III*::eGFP. (at) atrium. (vt) ventricle. **D** Graphic representation of eGFP chamber expression patterns of cohorts of embryos injected with *SMyHC III*::eGFP, *SMyHC IIIΔcNRE*::eGFP, and constructs containing point mutations in the cNRE Hexads A, B, and C (Mut A, B, and C, respectively) as well as a non-Hexad control mutation (Mut S). Embryos were analyzed at 48 h post-fertilization (hpf) and classified into three categories of cardiac expression patterns: exclusive atrium (at), exclusive ventricular (vt), and atrium plus ventricular (at+vt). chi-square test, ΔcNRE: **$p = 0.0026$; ****$p < 0.0001$, Mut A: ****$p < 0.0001$, Mut B: ***$p = 0.0002$; ****$p < 0.0001$, Mut C: ***$p = 0.0005$, Mut S: *$p = 0.0388$ (vt);

*$p = 0.0116$ (at+vt), comparing *SMyHC III* to each mutation and condition. **E** Frontal views, anterior is to the top, mouse embryos. *SMyHC III*::HAP (line 5, $n = 18$) isolated heart at 10.5 days post coitum (dpc), showing intense, dark blue, atrial coloring indicative of conspicuous HAP expression. **F** HAP assays in homogenates of atrial and non-atrial cardiac tissues in *SMyHC III*::HAP ($n = 18$), unpaired $t$ test, $p < 0.0001$. **G** *SMyHC IIIΔcNRE*::HAP (line 110, $n = 16$) isolated heart at 10.5 dpc, showing absence of HAP expression. **H** HAP assays in homogenates of atrial and non-atrial cardiac tissues in the *SMyHC IIIΔcNRE*::HAP mutant ($n = 16$), unpaired $t$ test, $p = 0.0013$. **I** Confocal image in lateral view, anterior is to the left, of a representative zebrafish embryo. Exclusive ventricular eGFP expression is observed at 48 hpf when injected with the *vmhc* promoter. **J** Confocal image in lateral view, anterior is to the left, of a representative zebrafish embryo. Expression of eGFP is detectable in both heart chambers at 48 hpf when injected with the *5xcNRE-vmhc* construct. **K** Graphical analysis of chamber expression patterns of the cohort of embryos injected with *vmhc* or *5xcNRE-vmhc* promoter constructs. (at) atrium. (vt) ventricle. chi-square test, $p < 0.05$, comparing *vmhc*::eGFP and *5xcNRE-vmhc*::eGFP embryos for each condition. Scale bars are 30 μm.

compared to WT cNRE (Fig. 3A) and to the mutant with a complete deletion of the cNRE (Fig. 3B, B′, B″). Mutation of Hexad A (Mut A) initially described by Wang et al.[6], was obtained by substituting the Hexad A sequence 5′-AGGACA-3′ for 5′-GTCGAC-3′ (Figs. 2A, B, 3C). Since the first two nucleotides are critical for nuclear receptor binding[14–16], dinucleotide substitutions were introduced in the two first positions of Hexad B (5′-AG-3′ changed to 5′-TT-3′) and Hexad C (5′-GG-3′ changed to 5′-TT-3′) to obtain, respectively, Mut B and Mut C (Figs. 2C–N, 3D, E). One dinucleotide substitution in the spacer region between Hexads B and C was designed as a non-Hexad control mutation (Mut S) (Figs. 2O, P, 3F). In zebrafish embryos, when comparing Mut A to the WT *SMyHC III*::eGFP promoter (Figs. 1D, 3A, A′), we observed a significant increase in the proportion of embryos with eGFP-positive cells in both atrium and ventricle (from 52%, $n = 45$ of 86 embryos, in WT to 80%, $n = 70$ of 87 embryos, in Mut A, $p < 0.001$) and a significant decrease in the proportion of embryos with eGFP-positive cells exclusively in the atrium (from 38%, $n = 33$ of 86 embryos, in WT to 8%, $n = 7$ of 87 embryos, in Mut A, $p < 0.0001$) (Figs. 1D, 3C, C′). We hence concluded that, in zebrafish, Hexad A is required for ventricular repression. In contrast, in mouse embryos, Mut A (Figs. 2B, 3C″) had no effect on HAP expression when compared to WT *SMyHC III*::HAP (Figs. 1E, 3A″, Supplementary Fig. S4). Similar to the effect of Mut A, for Mut B in zebrafish, we observed a significant increase in the proportion of eGFP-positive cells in both chambers (from 52%, $n = 45$ out of 86 embryos, in WT to 85%, $n = 62$ out of 73 embryos, in Mut B, $p < 0.0001$) and a significant decrease in the proportion of embryos with eGFP-positive cells exclusively in the atrium (from 38%, $n = 33$ out of 86 embryos, in WT to 12%, $n = 9$ out of 73 embryos, in Mut B, $p = 0.0002$) (Fig. 1D). For Mut B, we found a similar effect in mouse embryos, with a conspicuous release of HAP expression in the left ventricle and proximal outflow tract (Fig. 2C–H). Altogether, the experiments in both zebrafish and mice support the notion that Mut B released ventricular repression (Fig. 3D, D′, D″), thus suggesting that Hexad A together with Hexad B are required for strong repression of gene expression in the ventricle[5,7,8,17,18].

### Hexad C contains an atrial activator of the *SMyHC III*
Mutation of Hexad C (Mut C) resulted in a significant increase of ventricular-specific expression in zebrafish embryos (33%, $n = 16$ of 48 embryos, $p = 0.0005$) when compared to the *SMyHC III* control (9%, $n = 8$ of 86 embryos) (Fig. 1D). This increase was accompanied by a decrease of the proportion of embryos with eGFP expression only in the atrium, or in both the atrial and ventricular chambers (25%, $n = 12$, and 42%, $n = 20$, respectively of 48 embryos). Consistent with this result in zebrafish, we found decrease in atrial reporter expression in mouse Mut C mutants (Fig. 2I, J, M) relative to WT mice (Fig. 2K). Reduction of reporter expression in the atria is especially strong in the hearts of 10.5 dpc Mut C

mutant mice (Fig. 2M, N). Taken together, the reduction of atrial reporter expression of Mut C in both zebrafish and mouse (Fig. 3E, E′, E″) indicates that Hexad C is an atrial activator.

### Control mutations and integration of results
As a control for Hexad deletions, we assessed the effect of a mutation in a spacer region outside the Hexads (Mut S). This mutation is located between Hexads B and C, adjacent to Hexad B (Figs. 2O, 3F). In zebrafish, Mut S did not trigger a significant change in atrial-specific expression (47%, $n = 16$ of 34 embryos). However, we observed an increase of eGFP-positive cells in the ventricle (23%, $n = 8$ of 34 embryos) compared to the *SMyHC III* control (9%, $n = 8$ of 86 embryos) at the expense of eGFP expression in both chambers (29%, $n = 10$, of 34 embryos $p = 0.0388$) compared to the *SMyHC III* control (52%, $n = 45$ of 86 embryos) (Fig. 1D). This suggests that Mut S might contribute to the repression of atrial expression in zebrafish (Fig. 3F, F′). However, these results could not be confirmed in mice where Mut S had no effect on HAP expression compared to the WT *SMyHC III*::HAP (Figs. 2O, P, 3F″).

Altogether, our results demonstrate that, mechanistically, Hexad C plays the role of a *SMyHC III* atrial activator (Fig. 3). Furthermore, the phenotype displayed by Mut A and Mut B transgenic embryos is consistent with the release of a highly stereotyped ventricular expression pattern, as has previously been suggested for the deletion of the complete VDRE/RARE motif (i.e., Hexad A plus Hexad B)[6,7]. It is important to recognize that our cardiac chamber expression assays in the zebrafish yielded less robust results than those previously obtained in the quail[6,7]. However, as we were able to generate multiple stable transgenic *SMyHC III*::eGFP lines driving robust preferential atrial expression in the zebrafish (Supplementary Fig. S5), we nonetheless postulate that the atrial preference of the *SMyHC III* promoter is conserved across vertebrates.

### Evolutionary origins of the *SMyHC III* gene
Given that the quail *SMyHC III* cNRE drives preferential atrial expression in different vertebrates, we sought to define the evolutionary origins of this regulatory element, as well as of its association with the *SMyHC III* gene. We found that the *SMyHC III* genes constitute a strongly supported clade of avian-specific slow myosins[5,19] that are absent from primitive avians, such as *Struthio camelus* and *Tinamus guttatus* (paleognaths). Interestingly, in addition to an absence from the genomes of outgroup species, such as *Aligator mississipiensis* and *Aligator sinensis* (crocodilians), we were also unable to identify *SMyHC III* genes in the genomes of passeriform birds (Fig. 4, Supplementary Data 2). These data are consistent with the hypothesis that the *SMyHC III* gene originated at the root of the radiation of galliform birds during the Cretaceous, between 60 and 70 million years ago[11,20].

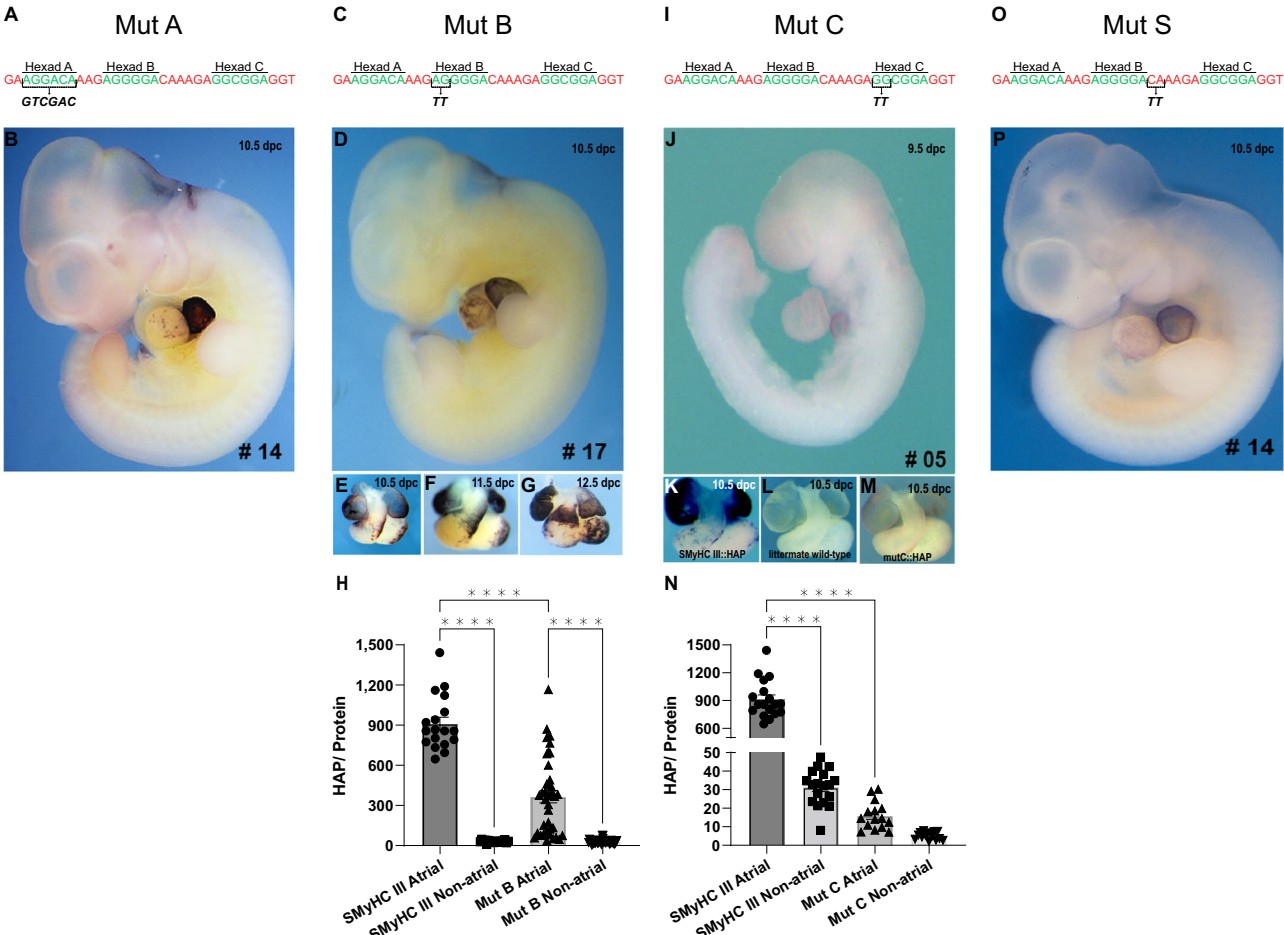

**Fig. 2 | Point mutations in Hexads B and C affect HAP expression in the mouse heart. A** Strategy for the mutation of Hexad A (Mut A). **B** Mouse line 14 (Mut A) at 10.5 days post coitum (dpc), showing atrial-specific expression of HAP. **C** Strategy for the mutation of Hexad B (Mut B). **D** Mouse line 17 (Mut B) at 10.5 dpc, showing expression of HAP. **E–G** Time course (10.5 dpc to 12.5 dpc) of cardiac expression in both chambers (atrium and ventricle) in mouse line 17 (Mut B). **H** Comparison of HAP assays in homogenates of atrial and non-atrial cardiac tissues from Mut B line 17 (n = 38) and *SMyHC III* line 5 (n = 18) in 10.5 dpc embryos, one-way ANOVA with Bonferroni post-hoc test, ****$p < 0.0001$. **I** Strategy for the

mutation of Hexad C (Mut C). **J** Mouse line 5 (Mut C) at 9.5 dpc. **K** Isolated heart from the *SMyHC III*::HAP line at 10.5 dpc. **L** Isolated heart from a wild-type littermate at 10.5 dpc. **M** Isolated heart from mouse line 5 (Mut C) at 10.5 dpc. **N** Comparison of HAP assays in homogenates of atrial and non-atrial cardiac tissues from Mut C line 5 (n = 15) and *SMyHC III* line 5 (n = 18) in 10.5 dpc embryos, one-way ANOVA with Bonferroni post-hoc test, ****$p < 0.0001$. **O** Strategy for the mutation of the non-Hexad control (Mut S) in the spacer sequence between Hexads B and C. **P** Representative mouse line 14 (Mut S) at 10.5 dpc, showing atrial-specific HAP expression.

## Evolutionary origins of the cNRE

Assaying 519 vertebrate genomes (at the exclusion of the genomes of galliform birds), we identified only 23 genomes containing cNRE-like signatures (Supplementary Data 3), demonstrating that cNRE-like DNA sequences are rarely found outside galliform birds. Focusing only on archosaurian (i.e., reptile and bird) genomes, we identified 55 cNRE-like sequences. Of these, 25 sequences (i.e., 45.5%) were characterized by 4 or more mismatches in the 32-bp stretch (representing a mismatch level of more than 12%). The other 30 sequences were significantly more conserved (Fig. 5, Supplementary Fig. S6) and included the cNREs associated with the 5′ and 3′ regions of the *SMyHC III* genes of galliform birds (Fig. 5, Supplementary Fig. S7). 5′ and 3′ flanking cNREs were thus identified in red-legged partridge (*Alectoris rufa*), wild turkey (*Meleagris gallopavo*), pinnated grouse (*Tympanuchus cupido*), greater sage-grouse (*Centrocercus urophasianus*), Gunnison sage-grouse (*Centrocercus minimus*), white-tailed ptarmigan (*Lagopus leucura*), Indian peafowl (*Pavo cristatus*), chicken (*Gallus gallus*), silver pheasant (*Lophura nycthemera*), brown eared pheasant (*Crossoptilon mantchuricum*), common pheasant (*Phasianus colchicus*), Mikado pheasant (*Syrmaticus mikado*), and helmeted guineafowl (*Numida meleagris*). Notably, the northern bobwhite (*Colinus virginianus*), the scaled quail (*Callipepla squamata*), the Chinese bamboo partridge (*Bambusicola

thoracicus*), and the Japanese quail (*Coturnix japonica*) have all lost their 3′ cNREs (Fig. 5, Supplementary Fig. S7). The genomic arrangement of single cNRE copies flanking the *SMyHC III* gene in 5′ and 3′ is thus likely the primitive configuration, as it was found in most galliform birds examined, as well as in *Numida meleagris*, an early branching galliform species. No cNRE-like sequences were found flanking *MYH6*, *MYH7* or any of the genes encoding myosins in the vertebrate genomes we studied, including those of crocodilians and paleognath birds. Taken together, the cNRE we define here, composed of the three Hexads A, B, and C, is specifically associated with galliform birds and originated early during galliform radiation[11,20].

## Are 5′ and 3′ *SMyHC III* cNREs remnants of ancient transposon insertions?

To gain insight into the origin of the cNRE as a genomic element, we assessed whether cNRE-like sequences are associated with a specific family of transposable elements in the target genomes[21]. When comparing the signature profiles of transposable elements in the regions flanking the cNRE-like sequences we identified in various vertebrate genomes (Supplementary Data 3), there was no systematic association of the cNRE-like sequences with any transposable element family (Supplementary Fig. S8). The same, negative, result was obtained when analyzing the cNRE-like

**Fig. 3 | Comparison of cNRE mutations between zebrafish and mice. A** *SMyHC III* promoter in (**A′**) zebrafish and (**A″**) mice. **B** cNRE deletion in (**B′**) zebrafish and (**B″**) mice. **C** Mutation of Hexad A (Mut A) in (**C′**) zebrafish and (**C″**) mice. **D** Mutation of Hexad B (Mut B) in (**D′**) zebrafish and (**D″**) mice. **E** Mutation of Hexad C (Mut C) in (**E′**) zebrafish and (**E″**) mice. **F** Mutation of the spacer sequence between Hexads B and C (Mut S) in (**F′**) zebrafish and (**F″**) mice. Zebrafish hearts in lateral views and mouse hearts in frontal view. (at) atrium. (vt) ventricle. Illustration created with BioRender.com.

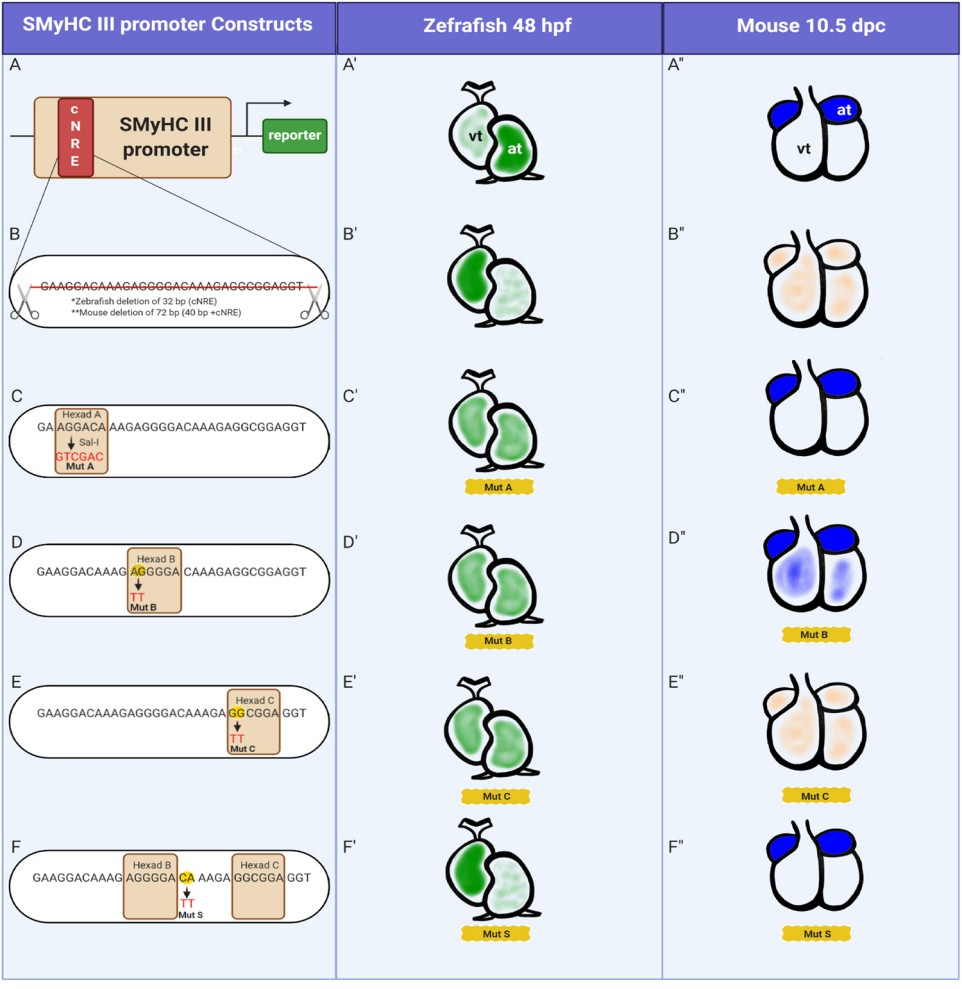

sequences identified in archosaurian genomes (Supplementary Fig. S9). To increase the resolution of our study, we next performed repeat masking analyses at the *SMyHC III* locus of the galliform birds: *Numida melleagris*, *Gallus gallus*, and *Coturnix coturnix*. Masking of repetitive sequences showed no overlap with the genomic location of the cNRE (Fig. 6) and thus supported the notion that the cNRE is a rare motif, distinct from low complexity sequences, satellites, simple repeat elements (SRes), and matrix attachment regions (Mars). Interestingly, repeat masking identified transposon-like sequences (i.e., ERVL and CR1) in the 3′ region of the *Gallus gallus SMyHC III* gene. However, sequence analyses revealed that the cNRE does not overlap the CR1 transposon remnant in the 3′ region of the *Gallus gallus SMyHC III* gene (Fig. 6). Taken together, the absence of TE sequences in the vicinity of the 5′ cNRE and the lack of connection between the 3′ cNRE and the residual CR1 transposon in *Gallus gallus* support the notion that the cNREs of the *SMyHC III* locus did not originate from transposon insertions.

**Are cNREs byproducts of viral recombination?**

We found that *SMyHC III* genes tend to be encoded in telomeric regions (of chromosome 18 in *Numida melleagris* and of chromosome 19 in *Gallus gallus*, for example), and there is circumstantial evidence that cNRE origins might be linked to this telomeric genome environment of the *SMyHC III* gene. In vertebrate chromosomes, telomeric ends are densely covered with direct repeat sequences that resemble the hexameric units that we now propose as major cNRE elements. For instance, the consensus telomeric repeat sequence is a hexad, which reads 5′-GGGTTA-3 in the complementary strand and the sequences of the cNRE are 5′-AGGACA-3′, 5′-AGGGGA-3′, and 5′-GGCGGA-3′ for, respectively, Hexads A, B, and C.

Intriguingly, repeats of telomeric hexamers flank certain regions of viral genomes, including those of Marek's disease virus (MDV/GaHV2), an oncogenic Alphaherpesvirus of galliform birds that causes T-cell lymphomas and can integrate itself into the germline[22]. Since galliform birds are known to be highly susceptible to viral integration[22–26], we hypothesized that the cNRE might have a viral origin. To test this hypothesis, we searched for cNRE-like sequences in Alphaherpesviruses that infect galliform birds, such as GaHV1/ILTV, the virus responsible for avian infectious laryngotracheitis in chicken, GaHV2/MDV1, GaHV3/MDV2, a non-pathogenic virus, and MeHV1/HTV, a non-pathogenic Meleagrid herpesvirus 1. The human herpes simplex virus 1 (HSV1) was included as an outgroup[27,28].

We found one cNRE-like sequences in GaHV1 (Table 1), providing proof of principle that galliform-specific viruses contain sequences that are similar to the cNREs of the *SMyHC III* locus and thus can, in principle, contribute cNREs to avian genomes via recombination. We further recovered six hits for cNRE-like sequences in HSV1, four in terminal repeat regions and two in unique sequence regions of the virus (Table 1). cNRE signatures are thus present in the sequences of Alphaherpesvirinae, suggesting that cNREs may be ancestral sequences from a family of viruses that infects and integrates into the genomes of vertebrates as diverse as avians and mammals. Since galliform birds are susceptible to infection by a host of other viruses besides Alphaherpesvirus, we screened for cNRE-like sequences in the avian viral database. In addition to herpesviruses, we found statistically significant cNRE-like matches in the genomes of papillomaviruses, paramyxoviruses, and retroviruses (Fig. 7). Of these, only paramyxovirus seem to lack the ability to integrate into avian genomes[29]. This suggests that cNREs might have originally been associated with the *SMyHC III* gene through viral infection of an ancestral galliform bird host

**Fig. 4 | SMyHC III is part of a strongly supported subfamily of galliform-specific myosins.** Phylogenetic analysis of the Myosin Heavy Chain (MYH) 6, 7, and 7B families in representative species of archosaurians. Maximum Likelihood tree of archosaurian MYH6, MYH7, and MYH7B proteins with branch support (Bootstrap percentages/Bayesian posterior probabilities) indicated at each node. "--" indicates that the node is not supported by Bayesian Inference. The tree is midpoint rooted and branch lengths correspond to sequence substitution rates.

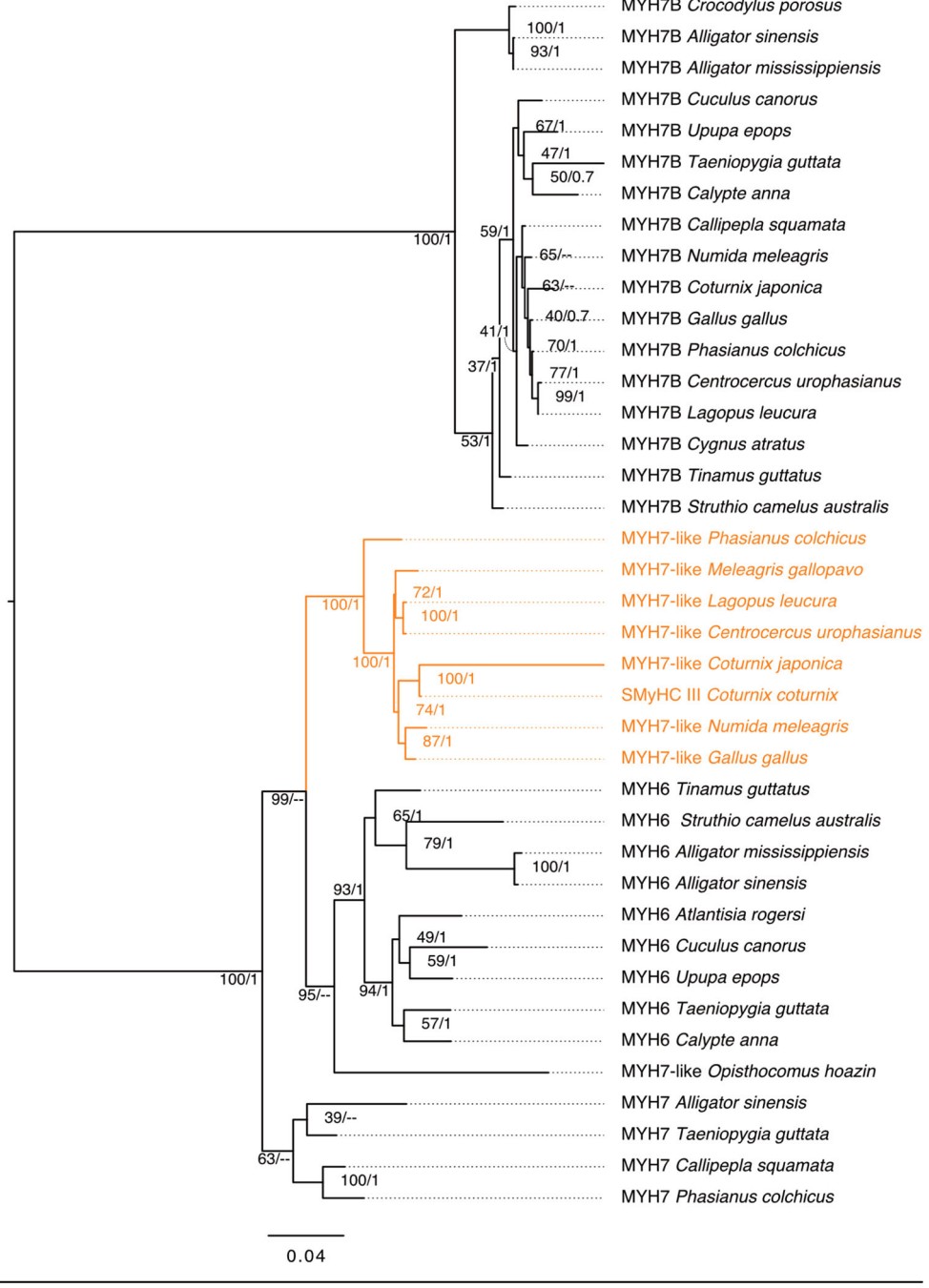

and that integration into the genome occurred early during the galliform radiation in the Cretaceous period, between 60 and 70 million years ago[11,20].

Taken together, our study demonstrates that the cNRE is a tripartite *cis*-regulatory element that confers atrial chamber preference and suggests that transcriptional modules of selective cardiac chamber expression emerged during early diversification of galliform birds, possibly by recombination between the host germline and a sequence of viral origin.

## Discussion

Cardiac development in vertebrates, i.e., the formation of a chambered heart, entails complex fate changes as well as morphogenetic movements and is controlled by an intricate gene regulatory network[1,30]. In this work, we have identified the cNRE, a new, 32-bp-long regulatory element located within the quail Slow Myosin Heavy Chain III (*SMyHC III*) promoter. We further showed that the cNRE sways cardiac expression towards a model of preferential atrial activation in zebrafish and mice. The cNRE ability to bias

cardiac expression crosses species barriers (from quail to mice and zebrafish), highlighting the importance of this novel element for understanding the evolution of gene regulation underlying the specification of cardiac chambers during early vertebrate development. Wang et al.[5,7] previously demonstrated that the VDRE/RARE element, which includes Hexads A and B of the cNRE, is responsible for ventricular repression of *SMyHC III* promoter activity in avian cardiac cells. They also identified a GATA-binding element in the *SMyHC III* promoter involved in activating expression in both the atrium and the ventricle. Missing from this detailed view of promoter function was the identity of the DNA sequence driving expression in atrial cells. By searching for potential novel nuclear receptor binding sites, we defined the cNRE as a 3′ expansion of the initial 17-bp long motif, adding a third Hexad. The 32-bp cNRE thus contains three Hexads (A, B, and C) and is responsible for preferential activity of the promoter in the atrium[5–8]. To functionally characterize the activity of the cNRE in the heart, we took advantage of the power of transient expression assays in

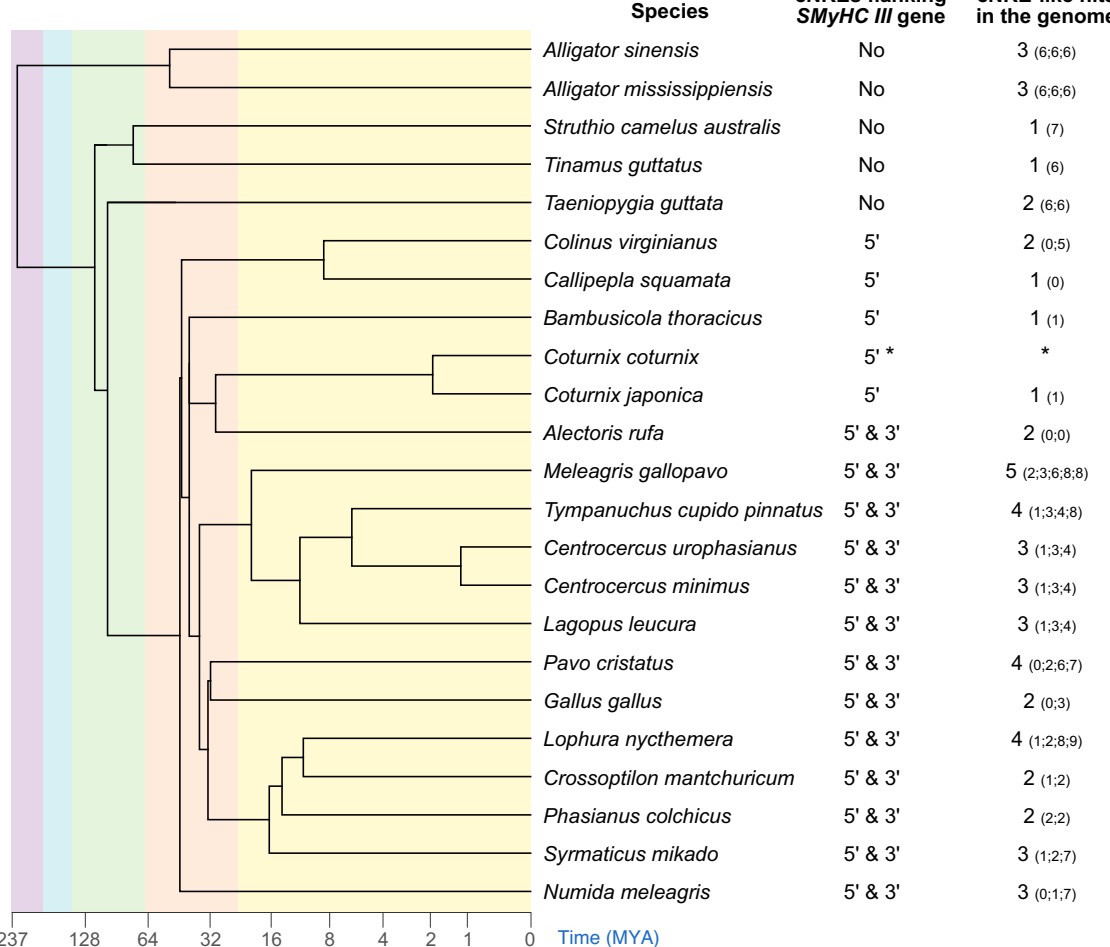

| Species | cNREs flanking *SMyHC III* gene | cNRE-like hits in the genome |
|---|---|---|
| *Alligator sinensis* | No | 3 (6;6;6) |
| *Alligator mississippiensis* | No | 3 (6;6;6) |
| *Struthio camelus australis* | No | 1 (7) |
| *Tinamus guttatus* | No | 1 (6) |
| *Taeniopygia guttata* | No | 2 (6;6) |
| *Colinus virginianus* | 5' | 2 (0;5) |
| *Callipepla squamata* | 5' | 1 (0) |
| *Bambusicola thoracicus* | 5' | 1 (1) |
| *Coturnix coturnix* | 5' * | * |
| *Coturnix japonica* | 5' | 1 (1) |
| *Alectoris rufa* | 5' & 3' | 2 (0;0) |
| *Meleagris gallopavo* | 5' & 3' | 5 (2;3;6;8;8) |
| *Tympanuchus cupido pinnatus* | 5' & 3' | 4 (1;3;4;8) |
| *Centrocercus urophasianus* | 5' & 3' | 3 (1;3;4) |
| *Centrocercus minimus* | 5' & 3' | 3 (1;3;4) |
| *Lagopus leucura* | 5' & 3' | 3 (1;3;4) |
| *Pavo cristatus* | 5' & 3' | 4 (0;2;6;7) |
| *Gallus gallus* | 5' & 3' | 2 (0;3) |
| *Lophura nycthemera* | 5' & 3' | 4 (1;2;8;9) |
| *Crossoptilon mantchuricum* | 5' & 3' | 2 (1;2) |
| *Phasianus colchicus* | 5' & 3' | 2 (2;2) |
| *Syrmaticus mikado* | 5' & 3' | 3 (1;2;7) |
| *Numida meleagris* | 5' & 3' | 3 (0;1;7) |

237   128   64   32   16   8   4   2   1   0    Time (MYA)

**Fig. 5 | cNREs flank the *SMyHC III* genes of galliform birds.** Phylogenetic tree of representative species of archosaurians, showing the presence (or absence) of cNREs in the 5′ and 3′ flanking regions of *SMyHC III* genes as well as the number of cNRE-like hits in the entire genome. Numbers in brackets represent the number of mismatches for each of the cNRE-like sequences relative to the *Coturnix coturnix* 5′ cNRE sequence. "*" indicates that the complete sequence of the *SMyHC III* locus is not available for *Coturnix coturnix*.

zebrafish embryos as well as of the generation of stable transgenic lines in mice[31]. Evaluation of the effects of point mutations in the cNRE in both zebrafish and mice ultimately allowed us to assess the role of each of the three Hexads constituting the cNRE. We found that the cNRE drives preferential reporter gene expression in the atrium of both zebrafish and mice, as deletion of the cNRE within the *SMyHC III* promoter abrogated the atrial-specific expression driven by this promoter.

Of note, while the activity of the wild-type *SMyHC III* promoter is almost completely limited to the atrium in mice, the promoter drives expression in both chambers of the zebrafish heart. It might be that the lineage-specific whole genome duplication in teleost fish[32] has secondarily altered the regulatory landscape controlling heart development[12]. Nevertheless, preferential atrial expression was significantly decreased in the zebrafish ΔcNRE mutant, and the multimerized cNRE construct still clearly shifted the activity of the zebrafish *vmhc* promoter in an atrial direction. We further sought to identify which regions in the cNRE sequence are critical to its activity and thus created constructs containing dinucleotide substitution mutations in the cNRE sequence for transient and stable transgenic analyses in zebrafish and mice, respectively. When mutating the sequences of Hexad A and B in zebrafish, we observed a decrease in isolated atrial expression concomitant with an increase in the number of individuals with simultaneous expression in both chambers. In mice, mutation of Hexad A did not affect expression, while mutation of Hexad B increased expression in ventricular chambers. These data results are consistent with previous in vitro studies performed in quail atrial cells, where removal of Hexads A and B led

to increased reporter gene expression in ventricular cells[8]. Taken together, these data suggest that Hexad B and potentially Hexad A act as ventricular repressors. Mutation of Hexad C resulted in a clear increase in ventricle-specific expression in zebrafish. We hypothesize that this effect is due to the specific loss of atrial expression in embryos that would normally be characterized by transgenic activity of the construct in both chambers. This notion is consistent with our finding that mutation of Hexad C in mice led to a pronounced decrease in atrial expression. We thus conclude that Hexad C plays an important role in atrial activation.

In summary, we demonstrate that the cNRE contains information needed for both atrial activation and ventricular repression of the *SMyHC III* promoter. Further work will be required to define the transcription factors binding to the cNRE sequence to regulate its activity. Although the *SMyHC III* promoter is not conserved among non-galliform vertebrate species, it is capable of driving atrium-specific expression in different animal models, including chicken, mouse, and zebrafish[5–9]. We postulate that the cNRE acts as a dual *cis*-regulatory module integrating both activating and repressing signals, some of which are likely mediated by members of the nuclear receptor superfamily via the VDRE/RARE binding sites[5,7,8].

It is difficult to amass direct evidence for the sequence of events culminating in the incorporation of two cNREs flanking *SMyHC III* genes at the telomeric region of a chromosome in the last common ancestor of galliform birds some 60 to 70 million years ago. Yet, we established that cNREs are not repetitive low-complexity elements, but rare sequences, occurring at less than one hundred hits in a diploid genome. In addition,

**Fig. 6 | cNREs are closely associated with the 5′ and 3′ ends of *SMyHC III* genes in galliform birds.** Schematics showing representative genomic location of cNREs and repeat sequences (obtained by RepeatMask analysis) near the *SMyHC III* genes (indicated in brown) in three galliform species: *Gallus gallus*, *Coturnix coturnix*, and *Numida meleagris*. The regions shown are *Gallus gallus* GalGal5.0 Chromosome 19: 15,000–37,000, *Coturnix coturnix* U53861 complete sequence, and *Numida meleagris* NumMel1.0 Chromosome 18: 22500-44500. The results show that, for each of the three species, one copy of the cNRE (red) is flanking the *SMyHC III* gene at both its 5′ and 3′ end. Note that the cNREs are distinct from satellite (Sat), simple repeat (SRe), low complexity (Low), and transposon sequences.

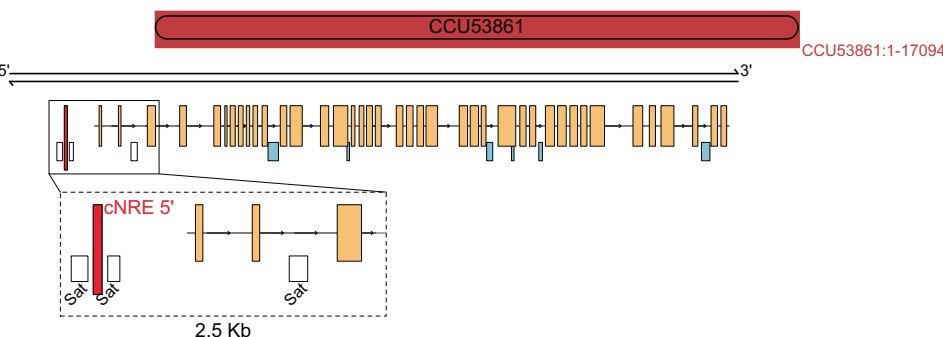

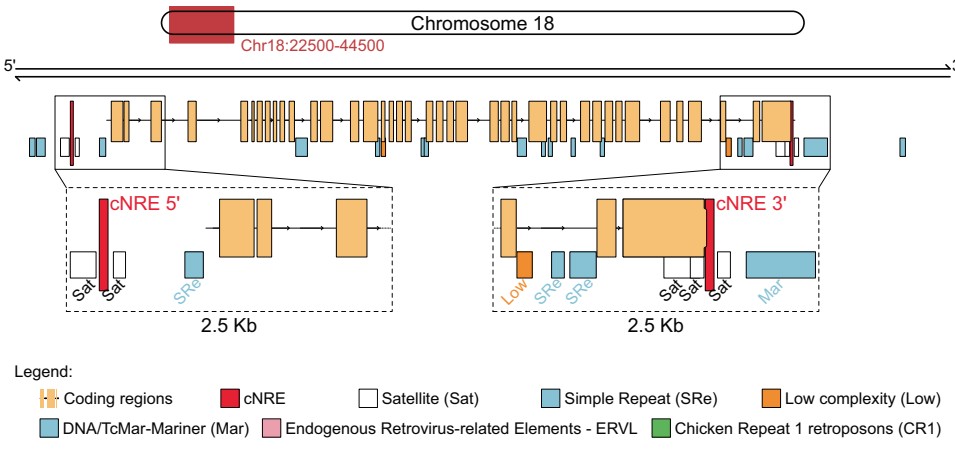

## Table 1 | cNRE-like hits in Alphaherpesvirus genomes

| Virus | Accession number | Hits | Start | End | Strand | Sequence | Mismatches |
|---|---|---|---|---|---|---|---|
| HSV1 | JN420341.1 | 6 | 4652 | 4683 | − | GAGGAAGAGGCAGAGGAGGAAGAGGCGGAGGC | 10 |
| | | | 8437 | 8469 | + | GAAGGAGAAGGAGGAGAGAGGGGGGGGGGAGAG | 10 |
| | | | 8727 | 8759 | + | AGGGGATCAAAGGGGGACAAAGAGGCGGGGGC | 9 |
| | | | 33929 | 33960 | − | CACGTACAAATCGGGGGCCATGAGGCCGCTGT | 11 |
| | | | 121375 | 121406 | + | GAGGAAGAGGCAGAGGAGGAAGAGGCGGAGGC | 10 |
| | | | 132264 | 132295 | + | GGCGGAGGAGGGGGGGGACGCGGGGGGCGGAGGA | 11 |
| GaHV1 (ILTV) | NC_006623.1 | 1 | 26529 | 26560 | + | GAACAGCGGCGAGACGAAAAAGAAGCGGAGGA | 11 |
| GaHV2 (MDV1) | AF243438.1 | 0 | | | | | − |
| GaHV3 (MDV2) | NC_002577.1 | 0 | | | | | − |
| MeHV1 (HTV) | AF291866.1 | 0 | | | | | − |

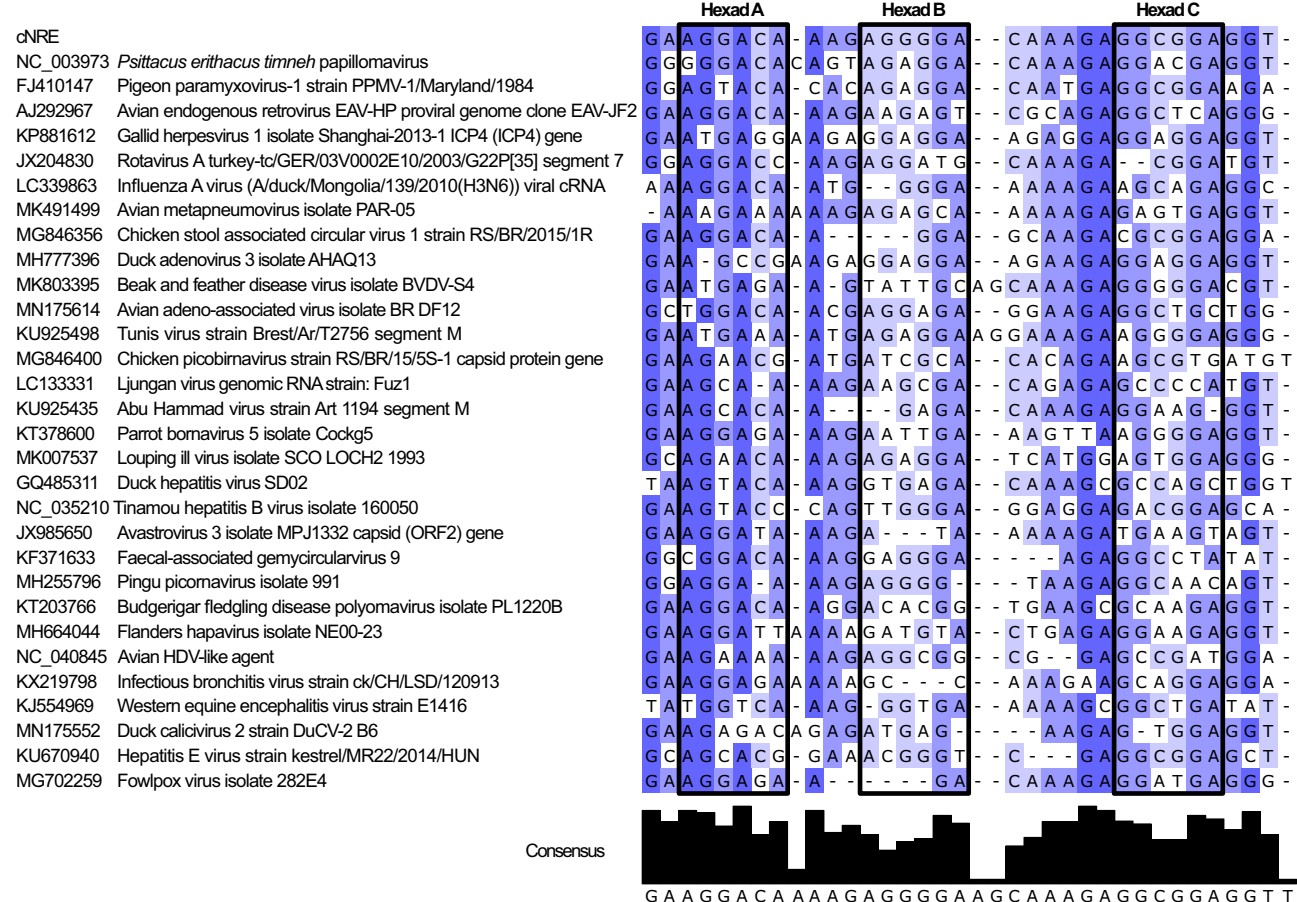

**Fig. 7 | cNRE-like sequences are found in viral genomes.** Alignment of the original cNRE with cNRE-like sequences identified in different viruses. Different shades of blue indicate the level of conservation of a given nucleotide in the alignment.

although containing similar hexameric modules, cNREs are not vertebrate telomeric repeats, and the origin of the cNRE cannot be linked directly to the activity of a particular family of transposable elements.

Since the primitive condition in galliform birds seems to be one *SMyHC III* gene flanked by 5′ and 3′ cNREs, it is likely that the appearance of *SMyHC III* in the avian genome and the emergence of the two flanking cNREs are mechanistically linked. We propose that an ancestral *SMyHC III* gene recombined into the germline at the telomeric region of a chromosome homologous to chromosome 18 of *Numida meleagris* and chromosome 19 of *Gallus gallus*, corresponding to Galliformes syntenic block 20 (Supplementary Fig. S10). This recombination event might have made use of a putative acceptor sequence of about 32-bp, which we dub here as a proto-cNRE, reminiscent of Lox-P (34-bp long) or FRT (34-bp long) binding sites for the products of cyclization-recombination genes, such as Cre or Flp recombinases, respectively[33]. The result of this event was the creation of 5′ and 3′ composite sites (the two flanking cNREs) that were formed by juxtaposition of the chromosomal acceptor sequences (the proto-cNRE) and nucleotides belonging to the extremities of the donor DNA containing the *SMyHC III* gene.

What could be the source of the cNRE? Galliform birds have been locked into a long co-evolutionary battle with viruses of the Alphaherpesviridae family, such as Marek's disease virus (MDV/GaHV2), a DNA virus capable of integrating its sequences into the genome of the host germline[22,26–28]. We found at least one cNRE-like signature in GaHV1 and six cNRE-like sequences in HSV1, suggesting that cNREs do reside in the sequence of Alphaherpesvirus. We thus propose that the acceptor proto-cNRE sequence emerged from an endogenous viral element[23], a genomic remnant of a recombination event caused by a viral infection of an ancestral host germline, perhaps through mechanisms of integration similar to those

of extant Alphaherpesviruses. According to this scenario, one or both cNRE sequences would have acquired the capacity to direct preferential atrial expression of the *SMyHC III* gene soon after the recombination event, bringing to bear the exapted potential of Hexads A, B, and C for ventricular repression and atrial activation.

## Limitations of our work

Our focus was on defining the physical limits and regulatory capabilities of the *cis*-regulatory module contained within −840 and −680 bp of the *SMyHC III* transcription start site, which contains the most powerful atrial stimulator in this gene, a seven-fold activator mapped in quail atrial cultures[5]. It was thus not possible to investigate all potential regulatory sequences of the *SMyHC III* gene, and we cannot formally exclude the possibility that there may be other relevant *SMyHC III* regulators, especially when considering the developmental constraints of different vertebrate taxa. The fact that point mutations of the cNRE result in different activities in mammals and teleost fish thus revealed important information about regulatory element evolution and highlighted potential limitations of cross-species assays. Taken together, we are thus convinced that our work demonstrated the general atrial bias of the cNRE in vertebrates and we argue that the differences observed quail, mice, and zebrafish might be the result of subtle, species-specific differences of DNA-protein binding affinities.

## Conclusions

Supported by in vivo experiments in zebrafish and mice, we establish that the quail cNRE, a sequence of merely 32-bp, carries information to control both atrial activation and ventricular repression. We further provide evidence for the evolution of this *cis*-regulatory element, ruling out an origin by transposable element integration and proposing a scenario for a potential

viral origin. Our work thus highlights the evolution of specific regulatory motifs in a sequence that is present exclusively in avian genomes but that can drive preferential atrial expression in different vertebrate taxa. In sum, this study sheds light on the origin of enhancers and defines the minimum amount of information required for regulating gene expression in an atrial-specific fashion.

## Materials and methods

### Bioinformatic profiling of nuclear receptor binding sites at the *SMyHC III* promoter

We devised a simulation approach to identify nuclear receptor binding sites (hexads) in the *SMyHC III* promoter. The principle of this approach is based on the Poisson–Boltzmann theory and aims at calculating interaction energies between nuclear receptors and DNA as an approximation of their respective binding affinities. Protein/DNA complexes were assembled for molecular dynamics profiling by positioning three-dimensional structures of nuclear receptor DNA binding domains on the three-dimensional structure of the cNRE DNA. RXR, RAR, and VDR crystal structures are available at the Protein Data Bank (http://www.rcsb.org) with the codes 1DSZ, 1KB4, and 1BY4, respectively[16,34,35]. The free binding energy was calculated for protein/DNA complexes using the trajectories obtained from molecular dynamics profiling with the software MM-PBSA in the AMBER package[36]. Each complex (AB) was split into two parts: the nuclear receptor structure (A) and the cNRE structure (B), and the energy was calculated for the whole complex (AB) as well as for each part, (A) and (B). Binding energy differences were obtained according to $\Delta\Delta G = \Delta G(AB) - (\Delta G(A) + \Delta G(B))$. Binding free energies for all cNRE hexads were plotted with reference to its first nucleotide. Data were pooled and analyzed as a box plot to identify values below the 10th percentile. These values were used to identify all potential hexads within the cNRE. To quantify the potential for nuclear receptor binding within the cNRE, we scored the number of times a given cNRE nucleotide was part of a hexad.

### COUPTF-II synthesis and purification

Human COUPTF-II (amino acids 74 through 414) in pET Sumo (Thermo Fisher Scientific) was expressed in *Escherichia coli* BL21 RP Codon Plus (Agilent). Bacteria were grown in LB broth (Thermo Fisher Scientific) until an $OD_{600nm}$ of 0.8 to 1.0 was reached and were then induced with 1 mM IPTG and 10 mM ZnCl at 16 °C for 16 h. After induction, the cultures were centrifuged (at 6000 rpm and 4 °C for 15 min), and the pellet was subsequently resuspended in 100 mM Tris-HCl, pH7.5, 300 mM NaCl, 10% glycerol, 0.1% triton X-100, and 2 mM β-mercaptoethanol. Cells were lysed by incubation with 100 mg lysozyme for 1 h on ice and sonication. Extracts were centrifuged at 15000 rpm and 4 °C for 40 min. Purification was performed by incubating extract with Talon resin (Clontech) at 4 °C for 1 h under agitation, with 1 mL of resin per liter of culture. After incubation, resin was washed in 100 mM Tris-HCl pH7.5, 300 mM NaCl, 10% glycerol, 0.1% triton X-100, 10 mM imidazole, and 2 mM β-mercaptoethanol for 20 column volumes, and protein was eluted in 100 mM Tris-HCl pH7.5, 300 mM NaCl, 10% glycerol, 0.1% triton X-100, 500 mM imidazole, and 2 mM β-mercaptoethanol for 10 column volumes. All steps were performed keeping extract, protein, and solutions at 4 °C. After purification, the SUMO protein tag was removed with SUMO protease (Thermo Fisher Scientific) (1:20 ratio of protease to COUPTF-II) at 4 °C for 18 h during dialysis. Tags were removed by incubation of cleaved protein in Talon resin for 1 h. Protein quality was checked by analytical gel filtration, SDS-PAGE electrophoresis, dynamic light scattering (DLS), and circular dichroism.

### GR synthesis and purification

A pre-inoculum was prepared from a fresh culture plate of the *E. coli* BL21 RP Codon Plus (Agilent) transformed with plasmid DNA containing GR expression vector in 120 mL of LB broth (Thermo Fisher Scientific) containing 50 μg/mL of kanamycin (Sigma-Aldrich) and incubated for 13 h at 37 °C and 200 rpm. 1% of the pre-inoculum was inoculated into culture medium in the presence of the same antibiotic. This new suspension was

maintained at 37 °C and 200 rpm until reaching an $OD_{600nm}$ of 0.8. The temperature was then reduced to 18 °C for 1 h, and protein expression was induced by adding 0.2 mM IPTG, 10 μM zinc acetate, and 10 μM dexamethasone. The suspension was incubated under shaking (200 rpm) at 18 °C for 16 h. Thereafter, the suspension was centrifuged at 7000 rpm and 4 °C for 15 min. Purification was performed first by affinity, as described for COUPTF-II, and subsequently by ion exchange chromatography. For this, the affinity-purified sample was diluted twice in the ion exchange equilibrium buffer and subjected to a double ion exchange: initially, the sample was loaded onto an anion exchange chromatography column using a Tricorn 10/100 Macro-Prep High Q (Bio-Rad) resin and then on a cationic column using a Tricorn 10/100 Macro-Prep High S (Bio-Rad) resin. The cationic column was washed with the ion exchange equilibrium buffer for 10 column volumes, and sequential elutions were made with different concentrations of NaCl: 400 mM, 600 mM, 800 mM, and 1 M. 36 samples of the chromatography fractions were evaluated by polyacrylamide gel electrophoresis, and the fractions containing the GR protein were collected. The purified GR protein was subsequently concentrated and cleaved as described for COUPTF-II, and the NaCl concentration was reduced to 150 mM through dilution and concentration using an Amicon Millipore filter (Merck) with a cutoff of 10 kDa. Buffer exchange was accomplished by an extra gel filtration step.

### Binding measurements

Single-stranded sense oligonucleotides were synthesized, coupled, at their 5′-ends, with a fluorescein molecule, and purified by high-pressure liquid chromatography (HPLC) (Integrated DNA Technologies). Double-stranded oligonucleotides were produced by annealing with their respective complementary strands in a LightCycler 480 (Perkin-Elmer) at equimolar concentrations in water. The pairing procedure was performed by heating the oligonucleotides to 100 °C for 10 min and subsequent slow cooling to 25 °C. The following oligonucleotides were prepared: A: 5′-ATATGAAGGACAAAAT-3′, B: 5′-ATAGAGGGGACAAAAT-3′, C: 5′-AGAGGCGGAGGTGGG-3′, AB: 5′-ATGAAGGACAAAGAGGGGA-CAAA-3′, BC: 5′-ATAGAGGGGACAAAGAGGCGGAGGT-3′, AC: 5′-ATGAAGGACAAAGAGGCGGAGGTG-3′, ABC: 5′-ATGAAGGACA AAGAGGGGACAAAGAGGCGGAGGT-3′. Anisotropy binding titration assays were performed using ClarioStar (BMG Labtech). Excitation was set to 480 nm and emission to 520 nm at 25 °C, with appropriate polarizers and filters. Anisotropy values were calculated according to Figueira et al.[37] and Fattori et al.[38]. Fluorescence anisotropy data represent averages of at least three independent experiments, performed with different protein batches and on different days, and anisotropy was measured until absolute errors were less than 5%. Binding affinities of human COUPTF-II and GR for different Hexads and Hexad combinations (A, B, C, AB, BC, AC, and ABC) were measured by serial dilution from 2 mM to 2 nM of protein in 20 mM Tris-HCL pH 7.5, 150 mM NaCl, 10% glycerol, 1 mM DTT, 10 μM zinc acetate, and 20 nM double-stranded, fluorescein-labeled oligonucleotides. Fluorescence curves were fitted using Origin (version 8.0) (OriginLab) by applying the Levenberg–Marquardt algorithm for fitting curves to nonlinear equations to determine binding affinity, dissociation constant ($K_d$), and Hill coefficient.

### Generation of mouse reporter lines

We generated mice containing mutations in critical nucleotides of Hexads A, B, and C (Mut A, B, and C, respectively) as well as a non-Hexad control mutation (Mut S). The constructs were synthesized with Agilent Quik-Change II XL Site-Directed Mutagenesis Kit following the manufacturer's instructions and using the primers listed in Table 2. Constructs were subsequently sequenced to confirm the substitutions. The Hexad A mutation (Mut A) was generated by mutagenesis of this Hexad's nucleotides to a SalI restriction enzyme site (5′-AGGACA-3′ to 5′-GTCGAC-3′). The mutations in Hexad B (Mut B) are point mutations of the two first nucleotides of this Hexad (5′-AGGGGA-3′ to 5′-TTGGGA-3′). Similarly, the mutation of Hexad C (Mut C) consisted of point mutations of the two first nucleotides of

**Table 2 | Oligonucleotides (in sense and antisense) used for hexad mutations**

| Hexad mutation | Oligonucleotide (sense) | Oligonucleotide (antisense) |
|---|---|---|
| Mut A | 5′-gaGTCGACaagaggggacaaagaggcggaggt-3′ | 5′-acctccgcctctttgtcccctcttGTCGACtc-3′ |
| Mut B | 5′-cttgcgaaggacaaagTTgggacaaagaggcggag-3′ | 5′-ctccgcctctttgtcccAActttgtccttcgcaag-3′ |
| Mut C | 5′-aggggacaaagaTTcggaggtggggctgg-3′ | 5′-ccagccccacctccgAAtctttgtccctc-3′ |
| Mut S | 5′-gaaggacaaagaggggaTTaagaggcggaggt-3′ | 5′-acctccgcctcttAAtccctctttgtccttc-3′ |

this Hexad (5′-GGCGGA-3′ to 5′-TTCGGA-3′). The non-Hexad control mutation (Mut S) was obtained by mutating two nucleotides in the putative nucleotide spacer between Hexads B and C (5′-AGGGGAcaaaga GGCGGA-3′ to 5′-AGGGGAttaagaGGCGGA-3′). *SMyHC III*::HAP transgenic lines 1 and 5 have previously been described[9], and four new *SMyHC III*::HAP transgenic lines were generated (2, 6, 27, and 29). Three *mutB*::HAP (5, 17, and 19) and seven *mutC*::HAP (3, 5, 7, 14, 15, 23, and 24) transgenic lines were established.

## Human alkaline phosphatase staining and histology
For HAP staining and paraffin sections, embryos and hearts were handled as described in ref. 9. For HAP assays, tissues were homogenized in a lysis buffer containing 0.2% Triton X-100 with a mini bouncer. HAP activity in cardiac tissues was measured with Phospha-Light, a chemiluminescent assay from PerkinElmer. HAP activity was normalized relative to protein concentration.

## Zebrafish wild-type and stable transgenic Tg(*vmhc*::mCherry) lines
To generate the Tg(*vmhc*::mCherry) line, we used the complete promoter upstream of the *vmhc* gene that drives powerful ventricle-specific expression in zebrafish[12,13]. The PCR-amplified fragment was cloned into pT2AL200R150G (courtesy of Dr. Koichi Kawakami), using the XhoI and HindIII restriction enzyme sites. The eGFP, between the ClaI and BamHI restriction enzyme sites, was substituted by mCherry. Injected embryos were raised to adulthood and an F2 generation was established. To generate the *Tol2-SMyHC III*::eGFP construct, the 840-bp upstream regulatory sequence of quail *SMyHC III* was excised at the SmaI and HindIII restriction enzyme sites from the *SMyHC III* pGL3 plasmid and cloned into pT2AL200R150G using the XhoI and HindIII restriction enzyme sites. Deletion of the 32-bp cNRE from the *SMyHC III* promoter (*SMyHC III*::Δ*cNRE*) was obtained by digesting the pGL3 plasmid with XhoI and HindIII and the subsequent cloning into pT2AL200R150G at the SmaI and HindIII restriction enzyme sites. For the mutational analyses of the *SMyHC III* promoter, specific mutations of the *Tol2-SMyHC III*::eGFP plasmid were generated using the Agilent QuikChange II XL Site-Directed Mutagenesis Kit following the manufacturer's instructions. Primers are listed in Table 2. The Tol2 system has been intensively used for transient injection assays in zebrafish for reducing mosaicism and increasing sensitivity, especially when studying heart chamber-specific expression[13]. However, coherent with previously published experimental evidence[12,13], some of the *vmhc* reporter lines exhibited ectopic expression, for example in the craniofacial region (Supplementary Fig. S5).

To construct the chimeric promoter *5xcNRE-vmhc*, we cloned five tandem repeats of the cNRE into the XhoI restriction enzyme site of the *Tol2-vmhc*::eGFP vector. The 5xcNRE sequence was obtained by annealing the following two oligonucleotides: 5′-CTAGGAAGGACAAAGAGG GGACAAAGAGGCGGAGGTGAAGGACAAAGAGGGGACAAAGAG GCGGAGGTGAAGGACAAAGAGGGGACAAAGAGGCGGAGCTGA AGGACAAAGAGGGGACAAAGAGGCGGAGGTGAAGGACAAAGA GGGGACAAAGAGGCGGAGGTCTCGAGA-3′ (sense) and 5′- GATC TCTCGAGACCTCCGCCTCTTTGTCCCCTCTTTGTCCTTCACCTC CGCCTCTTTGTCCCCTCTTTGTCCTTCACCTCCGCCTCTTTGTCC CCTCTTTGTCCTTCACCTCCGCCTCTTTGTCCCCTCTTTGTCCTT CACCTCCGCCTCTTTGTCCCCTCTTTGTCCTTC-3′ (antisense). For transient transgenic assays, all *Tol2*-based constructs were co-injected (~1 nL) into the cytoplasm of one-cell stage zebrafish embryos with

transposase mRNA, which was transcribed from the pCS-TP vector using the mMESSAGE mMACHINE SP6 Kit (Ambion). The master mix for injections was freshly prepared with 125 ng of the plasmid of interest, 175 ng of Tol2 transposase mRNA, 1 μL of 0.5% phenol red, and water to complete the final volume to 5 μL[39]. All constructs were microinjected in at least two independent experiments. Zebrafish embryos were staged and maintained at 28.5 °C, as previously described[40], and subsequently analyzed at 48 hpf. Transient expression of eGFP driven by the reporter constructs was scored in developing atria and ventricles of the heart, in the outflow tract of the heart as well as in non-cardiac tissues (Supplementary Fig. S3). Expression of eGFP in the atrium or the ventricle was scored as, respectively, atrium- or ventricle-specific expression, even if only a single eGFP-positive cell was detectable. Expression in both the atrium and the ventricle following the same criterion was thus scored as atrial and ventricular expression. eGFP expression in the outflow tract was classified as cardiac, but not part of the atrium or the ventricle. eGFP expression in any other tissue was classified as non-cardiac.

## Image analyses and processing
All imaging analyses were performed under a Nikon SMZ25 fluorescent stereomicroscope, and confocal imaging was carried out using a Leica SP8 microscope.

## Statistics and reproducibility
We used either the chi-square test, unpaired *t* test (non-parametric when appropriate) or one-way analysis of variance (ANOVA) with Bonferroni post-hoc test, and, for all analyses, a 95% confidence value was used to assess significance ($p < 0.05$). Data output is in column graphs with standard deviation, which was obtained using GraphPad Prism 6. Each experimental group was compared to its respective control group. All zebrafish embryos were microinjected in at least two independent experiments for each construct. The number of embryos analyzed is described in each graph. For HAP staining of *SMyHC III*::HAP transgenic lines 18 embryo hearts were analyzed, for *mutB*::HAP 38 embryo hearts were analyzed, for *mutC*::HAP 15 embryo hearts were analyzed, and *SMyHC IIIΔcNRE*::HAP 15 embryo hearts were analyzed.

## Phylogenetic analyses
Full-length amino acid sequences of myosin heavy chain (MYH) 6, 7, and 7B of representative Archosaurian species were recovered from publicly available sequence databases. Accession numbers are provided in Supplementary Data 3. An initial amino acid alignment was performed using MAFFT[41], which was followed by automated refinement using BMGE[42], and final manual curation. The phylogenetic tree was calculated based on the alignment of 39 sequences using 1935 amino acid positions and allowing gaps (Supplementary Data 4). Phylogenetic relationships were assessed using the Maximum Likelihood (ML) method as implemented in RAxML v8.0.0[43] and Bayesian Inference (BI) using Mr. Bayes v3.2.7[44]. The ML and BI trees were calculated applying a LG matrix[45]. The robustness of each node of the resulting ML tree was assessed by rapid bootstrap analyses (with 1000 pseudo replicates). Posterior probabilities for the BI tree were obtained in two Monte Carlo Markov Chain (MCMC) analyses run independently for 100,000 generations with trees sampled every 500 generations. The burn-in for this analysis was set at 0.25. Consensus trees were visualized, annotated, and midpoint rooted using Figtree v1.4 with branch lengths indicating the number of substitutions per site.

### Searching vertebrate genomes for cNRE-like sequences

To search different animal genomes for cNRE-like sequences, the command-line version of BLAST (BLAST+, version 2.5.0) was used. Global analyses of vertebrate genomes included all the taxa in the RefSeq Representative Genomes database, excluding the Galliformes (taxid:8976). For the cNRE searches of Archosauria genomes, we used all genomes available in the RefSeq Representative Genomes database as well as in the Whole-Genome Shotgun Contigs database. The searches were conducted using blastn-short, as this allowed for a search tailored to a query sequence as short as the cNRE. All parameters were set to default, except for word size (which was always set to 7 to maximize search accuracy) and eval (which was defined as 1). We retained, for further analysis, cNRE-like sequences that spanned all 3 Hexads (i.e., position 3 to 29) and that were characterized by a maximum of 1 mismatch per Hexad (multiple mismatches were allowed outside the Hexads) when compared to the *Coturnix coturnix* 5′ cNRE. Hit sequences against animal genomes were individually extracted from the databases, since BLAST+ is a local alignment algorithm that only yields parts of hit regions within a given genome. Each animal genome was thus loaded into R to obtain the exact position of the cNRE hit from the BLAST+ alignment. If the best hit for the cNRE was on the minus strand, the genome sequence was reverse complemented. Each sequence was subsequently extracted from the genome based on the genomic location of the initial BLAST+ hit and manually refined to allow a full alignment of the cNRE within a given genome sequence. In parallel, for each genome containing a cNRE-like sequence, the location of the best *Gallus gallus MYH7*-like/*SMyHC III* hit was determined by tblastn, using default parameters. The chromosome/scaffold for the best tblastn hit was compared to the genomic location of the cNRE. Only the genomes of Galliformes had best hits for both the cNRE (with less than 4 mismatches) and the *Gallus gallus MYH7*-like/*SMyHC III* on the same chromosome/scaffold. All extracted sequences were compiled into a single FASTA file and aligned using CLUSTAL OMEGA. The output alignment file was processed in Jalview to create the final alignment figure.

### Identification of transposable element signatures

To reveal the location of transposon elements in the vicinity of cNRE-like sequences, we isolated the genome sequences located between 5 kb and 6 kb upstream and between 5 kb and 6 kb downstream of each cNRE-like hit. Each genome sequence was then used as input for the search tool of the Dfam database (release 3.7), using Dfam curated threshold to estimate the significance of the hit[21]. The results were then compiled into individual figures to assess the distribution of transposable element family signatures within the genome sequences in relation to the cNRE-like hit. Additionally, RepeatMasker (http://www.repeatmasker.org) was used to screen for transposable elements and repetitive regions within the *SMyHC III* genome loci of *Numida melleagris* (NumMel1.0 NC_034426.1—Chromosome 18: 22,500–44,500) and *Gallus gallus* (GalGal5.0 NC_006106.4—Chromosome 19: 15,000-37,000) as well as within the available genomic sequence of the *Coturnix coturnix SMyHC III* locus (GenBank reference U53861). Results of the RMBlast were then loaded and visualized using Gviz[46] along with the *SMyHC III* coding sequences and the cNRE elements.

### Searching viral genomes

All available avian viruses in the NCBI virus database (as of February 19, 2020) were scanned for the cNRE sequence. For this, the R function pairwise alignment (in the Biostrings package, version 2.52.0) was used to create, for each virus, a local-global alignment score of the single best hit for the cNRE sequence and the cNRE reverse complement sequence (Supplementary Fig. S11). The "pattern" was set to the viral genome, the "subject" to the cNRE (or the cNRE reverse complement), the "type" to local-global, and all other settings were set to default. To calculate the p-value corresponding to a viral hit, 1000 32-bp-long random DNA sequences matching the cNRE length were generated and the pairwise alignment of each sequence was determined against each viral genome. The same set of random sequences was used to determine the p-value of every virus analyzed. If a cNRE

(or a cNRE reverse complement) hit in a viral genome had a local-global pairwise alignment score of more than 95% of the alignment score of the random sequences against this viral genome, this hit was considered as statistically significant. We retained the top 2,000 viral genomes, based on the pairwise alignment scores of the cNRE (or cNRE reverse complement). Of these, only the best-scoring virus of each viral family was chosen for further analysis. This measure ensured that no viral species or genus was overrepresented in the list of hits. Of the 2000 viral genomes, we obtained 27 unique virus families and 4 unclassified viral hits. The p-value of these 31 viruses was subsequently determined, with viruses returning a p-value of 0.05 or less being retained. These hit sequences of viral genomes were saved in a single FASTA file and the FASTA file was processed using CLUSTAL OMEGA to produce a multiple sequence alignment. If a hit in a virus genome was to the cNRE reverse complement, the viral hit sequence was reverse complemented before it was included in the FASTA file. The output file was then opened in Jalview and processed to highlight different levels of sequence conservation.

### Reporting summary

Further information on research design is available in the Nature Portfolio Reporting Summary linked to this article.

### Data availability

The datasets used and/or analyzed during the current study are available from the corresponding authors on reasonable request and available in the GitHub repository: https://github.com/Ramialison-Lab/cNRE-Genomics-Analysis.

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

## Acknowledgements

The authors thank Dr. Koichi Kawakami for providing the plasmid used to produce transgenic zebrafish. We thank the members of the Ramialison group and Akriti Varshney, Gulrez Chahal, and Julian Stolper for their feedback and support and Jeannette Hallab, Jeanette Rientjes, and Ekaterina Salimova for their assistance with experimental troubleshooting and design. We thank the members of the Monash Bioinformatics Platform for invaluable advice, especially Stuart Archer, Adele Barugahare, Paul Harrison, David Powell, Michael See, and Nick Wong. We thank the ARMI FishCore staff and Melbourne University Fish Facility. We also thank Lucas Buscaratti for his drawing created with BioRender.com which helped us to illustrate the different mutations on the *SMyHC III* promoter comparing mice and zebrafish. This work was supported in part by FAPESP grants 00/04082-1, 03/06555-2, 15/12549-2, and 18/09839-7 and by the Coordenação de Aperfeiçoamento de Pessoal de Nível Superior—Brasil (CAPES)—Finance Code 001, by CNPq grant 481983/2013-9, and by the Ceara State Scientist-in-chief program of Fundação Cearense de Apoio ao Desenvolvimento Científico e Tecnológico (FUNCAP 08908197/2019) to J.X.N. M.R. is supported by Grants from the Australian Research Council and the NHMRC. The Australian Regenerative Medicine Institute is supported by grants from the State Government of Victoria and the Australian Government. The Novo Nordisk Foundation Center for Stem Cell Medicine is supported by a Novo Foundation Grant (NNF21CC0073729). M.S. is funded by the CNRS and the ANR (ANR-21-CE34-0006-02). The authors wish to dedicate this work to the memories of our friends Gang Feng Wang, Ralff C. J. Ribeiro, and Esfir Slonimsky.

## Author contributions

L.N.S. generated transgenic zebrafish, contributed to discussions, and wrote this manuscript. A.M.S.C., P.R.J., and S.D. contributed to zebrafish transgenic analyses. M.N., J.E.C., and M.R. performed bioinformatics analyses, contributed to discussions, and co-wrote the manuscript. A.C.S., H.A.C., M.V., and T.G.F.M. carried out experiments on transgenic mice. M.B.G., H.T.N., and P.O. performed bioinformatics analyses. I.L.T. and A.C.M.F. performed biophysical assays. F.E.S., G.F.W., N.R., and W.N., contributed to

the conception of this study. M.S. contributed to the conception of this study, scientific discussions, and manuscript writing. J.X.N. was responsible for project design, mouse transgenics, scientific discussions, and manuscript writing. All authors have read and approved the final manuscript.

## Ethics approval and consent to participate
We confirm that all relevant ethical guidelines have been followed and that any necessary IRB and ethics committee approvals have been obtained. The present study was approved by the Ethics Committee on Animal Use, Institute of Biomedical Sciences, University of São Paulo (CEUA-ICB/USP), by the Ethics Committee on Animal Use, Brazilian Center for Research in Energy and Materials (CEUA-CNPEM) protocol number 24, by the University of Melbourne guidelines and local ethics committee, and by the Monash University Animal Ethics Committee.

## Competing interests
The authors declare no competing interests.

## Additional information

[1]Brazilian Biosciences National Laboratory (LNBio), Brazilian Center of Research in Energy and Materials (CNPEM), Campinas, SP, Brazil. [2]Australian Regenerative Medicine Institute, Monash University, VIC Australia - Systems Biology Institute, Melbourne, Australia. [3]Department of Cell and Developmental Biology, Institute of Biomedical Sciences, University of São Paulo (USP), São Paulo, SP, Brazil. [4]Laboratoire de Biologie du Développement de Villefranche-sur-Mer, Institut de la Mer de Villefranche, Sorbonne Université, CNRS, Villefranche-sur-Mer, France. [5]Faculdade Santa Marcelina - São Paulo, São Paulo, SP, Brazil. [6]Department of Medicine, Stanford University, Stanford, CA, USA. [7]School of BioSciences, University of Melbourne, Parkville, VIC, Australia. [8]The Jackson Laboratory, Bar Harbor, Maine, USA. [9]National Heart and Lung Institute, Imperial College London, London, UK. [10]Murdoch Children's Research Institute, Parkville, VIC, Australia. [11]Department of Morphology, Federal University of Ceará (UFC), Ceará, CE, Brazil. [12]Health Scientist-in-Chief of Ceará State, Fundação Cearense de Apoio ao Desenvolvimento Científico e Tecnológico, Ceará, CE, Brazil. [13]Deceased: Gang Feng Wang. ✉e-mail: mirana.ramialison@mcri.edu.au; josexavierneto@gmail.com

