## [Peer Review File · Communications Biology]

Reviewers' comments:

Reviewer #1 (Remarks to the Author):

Nunes Santos et al.

In this manuscript the authors revisit cis-element regulatory sequences of the quail Slow Myosin Heavy Chain III gene. In a series of studies from the mid-90s through around 2001, this region just upstream of the gene had been investigated by the Stockdale laboratory, using at the time cutting-edge “promoter bashing”. They showed convincingly, including using transgenic chick experiments, that a VDR/RXR element activates expression in the atrium, that a nearby GATA element is essential for cardiac expression, and that IRX4 is a negative regulator in the ventricle. This stood as a fine example for chamber specific gene regulation. It is not exactly clear why the authors felt the need to investigate this region in further detail, but an *in silico* analysis suggested to them that another nearby hexad sequence (called hexad C) might also be important, so they introduced mutations in this and other sequences of the “complex nuclear receptor element” or cNRE, and tested reporter expression in transient zebrafish and stable murine transgenics. They followed this with comparative genomics for the element across species and viruses and conclude that the regulatory element may have been derived by integration of an endogenous viral element. Therefore, the manuscript has two parts, one that examines several mutations, and a second in comparative genomics. Overall, the study provides little progress in understanding chamber-specific gene expression, and no significant mechanistic insight into regulation of the SMyHCIII gene.

1) The author’s claim that “Hexad C is the long sought atrial activator” is not supported by the evidence. Zebrafish have been used successfully for analyzing regulatory elements but in this case the data are not convincing. Based on supplemental data, 75% of injected embryos express the reporter in “other tissues”. Although not well documented (no data is shown) the reporter is not even cardiac specific, much less atrial specific. With the WT sequences over 60% of the embryos express the reporter in the ventricle, so clearly it is not behaving as an accurate atrial-specific reporter, either due to technical issues of transgenesis, or species differences. Some changes in the relative likelihood for the reporter to be expressed in one or both chambers are statistically significant, but the criteria for binning the results is unclear. For example, is one positive cell considered chamber expression? There is no evidence for function of the elements in binding specific factors.

2) The mouse transgenic assays are somewhat more convincing, perhaps either because of a closer evolutionary relationship, or because these are stable lines. Here, mutation of the Hexad C eliminates reporter activity. However, it does not distinguish reporter activity from chamber specificity, and it’s unclear why mutation of Hexad A, shown previously to be important in the chick, has no effect. Lack of expression for any one construct could simply be due to integration in a bad chromosomal site. It is also unclear exactly how many embryos were evaluated and how much variation was seen between lines.

3) The second part of the paper using comparative genomics may be of interest to evolutionary

biologists, but this reviewer is not competent to judge the impact.

Reviewer #2 (Remarks to the Author):

In the manuscript titled "Unraveling the evolutionary origin of the complex Nuclear Receptor Element (cNRE), a cis-regulatory module required for preferential expression in the atrial chamber", Santos and colleagues characterized a novel atrium specific activating elements known as cNRE, which comprises three Hexads sequences (A + B + C) located upstream of the SMyHC III promoter. In zebrafish and mice, cNRE drives preferential atrial expression of SMyHC III. Deletion of cNRE from SMyHC III promoter leads to a reduction in overall atrial expression while promoting ventricular-specific expression. The introduction of five tandem repeats of cNRE into the ventricular-specific vmhc promoter results in robust ventricular-specific expression with a shift towards atrial activation. Moreover, the authors elucidate the roles of the Hexad sequences within cNRE. Hexad A and Hexad B function as ventricular repressor elements, while Hexad C acts as an atrial activator for SMyHC III. Finally, the authors trace the evolutionary origins of cNRE and present evidence suggesting that it may originate from an endogenous viral element. Overall, this is a very impressive study that will contribute to a better understanding of the evolutionary pathway to achieve cardiac chamber-specific expression. However, some specific issues should be addressed before publication.

Major points:

To assess the sufficiency of cNRE for driving preferential atrial expression, the authors incorporated five tandem repeats of cNRE into the ventricular-specific vmhc promoter to examine its ability to bias ventricular-specific expression towards atrial activation. However, the results indicate that the vmhc promoter sequence appears to be somewhat shorter, as it did not exclusively drive vmhc expression in the ventricles. It is crucial to extend the vmhc promoter sequence to eliminate its atrial expression. The full 2.2 kb upstream of vmhc promoter may be necessary to ensure robust ventricular-specific expression (Zhang & Xu, 2009).

Minor points:

- 1) Figure 1F should be referenced within the manuscript.
- 2) The representation of dinucleotide substitutions (from '5'-GG-3' to '5'-TT-3') for Mut B and Mut C in the manuscript does not align with the labeling in Figure 2C and Figure 2I.

Reviewer #3 (Remarks to the Author):

1. What are the major claims of the paper?

This paper deals with cardiac gene regulatory networks. Specifically, the atrial-specific regulation of the Slow Myosin Heavy Chain III (SMYHC III) gene expression is examined. The authors claim to have discovered a third hexad sequence in the SMYHC III promoter that is responsible for atrial-specific gene regulation, and that this might have an evolutionary history dating back more than 60 million years ago, through a viral integration into an ancestral galliform bird host.

2. Are they novel and will they be of interest to others in the community and the wider field?

Yes - specifically for those interested in developmental biology of how optimal cardiac function is regulated. In broader terms, the idea of ancient viral integration and changes in gene regulatory networks is likely to continue to be a common theme for many different systems. It is interesting to note that these repeats appear in the telomere regions.

3. If the conclusions are not original, it would be helpful if you could provide relevant references.

I think the conclusions are original - there's certainly lots of other papers about ancient viral integration, but I think the authors build a compelling story for this case.

4. Is the work convincing, and if not, what further evidence would be required to strengthen the conclusions?

Yes, it's convincing, and described clearly.

5. On a more subjective note, do you feel that the paper will influence thinking in the field?

Potentially it could. The authors ask a reasonable question - obviously the gene regulatory networks must somehow allow specific regulation of contractile proteins that are different for each chamber of the heart - how does this happen? And they come up with a testable hypothesis - the regulation is similar to what's already known (a purine-rich region, with two known Hexads) and for SMYHC III regulation, a 32 bp repeat with three purine-rich hexanucleotide repeats is responsible for chamber-specific gene expression regulation.

6. Please feel free to raise any further questions and concerns about the paper.

No further questions.

7. We would also be grateful if you could comment on the appropriateness and validity of any statistical analysis, as well the ability of a researcher to reproduce the work, given the level of detail provided.

I have no problems with the current use of statistics in this manuscript.

Reviewer #1

In this manuscript the authors revisit cis-element regulatory sequences of the quail Slow Myosin Heavy Chain III gene. In a series of studies from the mid-90s through around 2001, this region just upstream of the gene had been investigated by the Stockdale laboratory, using at the time cutting-edge “promoter bashing”. They showed convincingly, including using transgenic chick experiments, that a VDR/RXR element activates expression in the atrium, that a nearby GATA element is essential for cardiac expression, and that IRX4 is a negative regulator in the ventricle. This stood as a fine example for chamber specific gene regulation. It is not exactly clear why the authors felt the need to investigate this region in further detail, but an *in silico* analysis suggested to them that another nearby hexad sequence (called hexad C) might also be important, so they introduced mutations in this and other sequences of the “complex nuclear receptor element” or cNRE, and tested reporter expression in transient zebrafish and stable murine transgenics. They followed this with comparative genomics for the element across species and viruses and conclude that the regulatory element may have been derived by integration of an endogenous viral element. Therefore, the manuscript has two parts, one that examines several mutations, and a second in comparative genomics. Overall, the study provides little progress in understanding chamber-specific gene expression, and no significant mechanistic insight into regulation of the SMyHCIII gene.

1) The author’s claim that “Hexad C is the long sought atrial activator” is not supported by the evidence. Zebrafish have been used successfully for analyzing regulatory elements but in this case the data are not convincing. Based on supplemental data, 75% of injected embryos express the reporter in “other tissues”. Although not well documented (no data is shown) the reporter is not even cardiac specific, much less atrial specific. With the WT sequences over 60% of the embryos express the reporter in the ventricle, so clearly it is not behaving as an accurate atrial-specific reporter, either due to technical issues of transgenesis, or species differences. Some changes in the relative likelihood for the reporter to be expressed in one or both chambers are statistically significant, but the criteria for binning the results is unclear. For example, is one positive cell considered chamber expression? There is no evidence for function of the elements in binding specific factors.

2) The mouse transgenic assays are somewhat more convincing, perhaps either because of a closer evolutionary relationship, or because these are stable lines. Here, mutation of the Hexad C eliminates reporter activity. However, it does not distinguish reporter activity from chamber specificity, and it’s unclear why mutation of Hexad A, shown previously to be important in the chick, has no effect. Lack of expression for any one construct could simply be due to integration in a bad chromosomal site. It is also unclear exactly how many embryos were evaluated and how much variation was seen between lines.

3) The second part of the paper using comparative genomics may be of interest to evolutionary biologists, but this reviewer is not competent to judge the impact.

Author's answers to reviewer's questions, comments, and criticisms

It is not exactly clear why the authors felt the need to investigate this region in further detail, but an *in silico* analysis suggested to them that another nearby hexad sequence (called hexad C) might also be important, so they introduced mutations in this and other sequences of the “complex nuclear receptor element” or cNRE, and tested reporter expression in transient zebrafish and stable murine transgenics.

Thank you very much for this comment. We felt that a further investigation of the SMyHC III promoter was warranted, because our previous work had established that the mechanisms of preferential atrial expression in mice were different from those established in quail (Masters dissertation of Matos, TGF 2002). We thus wanted to better understand the evolution of this regulatory element, including mechanistic differences in the regulation of preferential atrial expression. This work should be seen as the next step in this direction. And, as rightfully pointed out by this reviewer, we used several different aspects to address our biological question. We used functional analyses in model organisms to better characterize the regulatory motifs within the SMyHC III promoter, identifying the cNRE as an important component, and highlighting species-specific differences. In parallel, however, we set out to reconstruct the evolutionary origins of this regulatory element, an aspect of our work that has not been commented by this reviewer, but that has been very positively assessed by reviewer #3, indicating that our work should not only be judged by the level of novel insights we provide into the mechanisms of gene regulation.

Overall, the study provides little progress in understanding chamber-specific gene expression, and no significant mechanistic insight into regulation of the SMyHCIII gene.

We would like to respectfully disagree with this statement of the reviewer. Our identification of a novel, previously unidentified, evolutionarily-conserved, atrial-activating Hexad C with the cNRE is highly relevant for our understanding of how preferential atrial expression is organized in vertebrates. Therefore, mechanistically, our data shows that in addition to a general cardiac activator (a GATA factor) and a ventricular repressor, IRX4, SMyHC III-driven expression requires an atrial activator. However, in order to add additional mechanistic insights to our study, we included, in the revised manuscript, a new supplementary figure featuring a functional characterization of the binding affinities of cNRE Hexads A, B, and C to two different nuclear receptors, COUPTF-II (COUP transcription factor 2) and GR (glucocorticoid receptor). Using fluorescence anisotropy, we demonstrate specific binding to the cNRE of COUPTF-II, but not GR, highlighting that cNRE binding is restricted to a subset of nuclear receptors and suggesting that the cNRE acts as a nuclear receptor signaling hub during cardiac development.

The author's claim that “Hexad C is the long sought atrial activator” is not supported by the evidence.

The reviewer is correct to indicate that this phrase might be an overstatement, in particular in the comparative context presented in the manuscript. We have thus removed this statement from the revised version of the manuscript. However, please note that a comparison between atrial expression in wild type SMyHC III mouse transgenics and Hexad C mutants show a very substantial reduction in atrial activity driven by the mutated Hexad C, consistent with the interpretation that the role of the unmutated Hexad C is atrial activation.

Zebrafish have been used successfully for analyzing regulatory elements but in this case the data are not convincing. Based on supplemental data, 75% of injected embryos express the reporter in “other tissues”.

We would like to apologize to the reviewer if our statements concerning the scoring of reporter expression domains in the manuscript were not clear enough. It is common for promoter constructs to direct expression to several tissues, as has been described, for example, for the vmhc promoter, which is nonetheless considered as being highly specific for directing ventricular expression (Jin et al., 2009; Zhang & Xu, 2009). There is thus no contradiction between directing expression to a specific tissue within a given organ like the heart (the ventricle versus the atrium) and also being expressed elsewhere (like in the craniofacial region) (Jin et al., 2009; Zhang & Xu, 2009). To improve our dataset and avoid confusion, we reanalyzed our injected embryos for the revision and added details about our transgenic analyses, the scoring mechanisms, and potential off targets to the revised version of the manuscript.

*Jin, D., Ni, T.T., Hou, J., Rellinger, E., Zhong, T.P. (2009). Promoter analysis of ventricular myosin heavy chain (vmhc) in zebrafish embryos. **Dev. Dyn.** 238, 1760–1767. doi: 10.1002/dvdy.22000*

*Zhang, R., Xu, X. (2009). Transient and transgenic analysis of the zebrafish ventricular myosin heavy chain (vmhc) promoter: an inhibitory mechanism of ventricle-specific gene expression. **Dev. Dyn.** 238, 1564–1573. doi: 10.1002/dvdy.21929*

Although not well documented (no data is shown) the reporter is not even cardiac specific, much less atrial specific. With the WT sequences over 60% of the embryos express the reporter in the ventricle, so clearly it is not behaving as an accurate atrial-specific reporter, either due to technical issues of transgenesis, or species differences. Some changes in the relative likelihood for the reporter to be expressed in one or both chambers are statistically significant, but the criteria for binning the results is unclear. For example, is one positive cell considered chamber expression?

Please see comment above. We apologize for not having been explicit enough about these points in the original manuscript. We have added detailed information on the scoring criteria

of the transgenic experiments to the revised manuscript and have verified and specified the scores in the results section. Please note that representative information about the scoring has already been provided in Figure 1 and this is still the case in the revised manuscript. Furthermore, as this reviewer questions the specificity of our findings, we have added a new supplementary figure to the revised manuscript showing expression of SMyHCIII::eGFP and vmhc::mCherry in stable zebrafish transgenics, highlighting their specificity to, respectively, the atrium and the ventricle. This figure also shows expression in “other tissues” to further clarify this point in the revised manuscript.

There is no evidence for function of the elements in binding specific factors.

As discussed above, although we feel that the quality of our work should not only be judged by the amount of functional data we report, we have nonetheless added a new supplementary figure featuring a characterization of the binding affinities of cNRE Hexads A, B, and C to two nuclear receptors using anisotropy. In the revised manuscript, we now provide functional evidence that, in addition to the nuclear receptors VDR, RAR α , and RXR α , the cNRE can also be bound by COUP-TFII, but not GR. The cNRE thus mediates binding to several different nuclear receptors, which likely influences its regulatory activity during development.

The mouse transgenic assays are somewhat more convincing, perhaps either because of a closer evolutionary relationship, or because these are stable lines. Here, mutation of the Hexad C eliminates reporter activity. However, it does not distinguish reporter activity from chamber specificity, and it's unclear why mutation of Hexad A, shown previously to be important in the chick, has no effect. Lack of expression for any one construct could simply be due to integration in a bad chromosomal site. It is also unclear exactly how many embryos were evaluated and how much variation was seen between lines.

We thank the reviewer for this comment. However, it is important to clarify the situation regarding the phenotypes of transgenic mice harboring mutated Hexad A SMyHC III promoters. Seven independent mouse lines were produced, and none displayed increased ventricular expression as shown in quail embryos. We agree with reviewer #1 that bad chromosomal sites may block expression, but that is not what we saw. For the seven Hexad A mouse mutant lines produced, there was preferential atrial expression, just as in wild type mice transgenic for the wild type SMyHC III promoter. Thus, integration in unfavorable chromosomal sites cannot be the reason since one would expect these unfavorable integrations would not support any kind of expression. Yet, the reviewer's comments address a more fundamental issue, which is marked by the differences we see between in the activities of the reporter constructs in the two species we use in this study and the published record on experimental evidence in chicken. We have addressed this issue in the revised version by expanding the results section accordingly.

The second part of the paper using comparative genomics may be of interest to evolutionary biologists, but this reviewer is not competent to judge the impact.

—

Reviewer #2

In the manuscript titled "Unraveling the evolutionary origin of the complex Nuclear Receptor Element (cNRE), a cis-regulatory module required for preferential expression in the atrial chamber", Santos and colleagues characterized a novel atrium specific activating elements known as cNRE, which comprises three Hexads sequences (A + B + C) located upstream of the SMyHC III promoter. In zebrafish and mice, cNRE drives preferential atrial expression of SMyHC III. Deletion of cNRE from SMyHC III promoter leads to a reduction in overall atrial expression while promoting ventricular-specific expression. The introduction of five tandem repeats of cNRE into the ventricular-specific vmhc promoter results in robust ventricular-specific expression with a shift towards atrial activation. Moreover, the authors elucidate the roles of the Hexad sequences within cNRE. Hexad A and Hexad B function as ventricular repressor elements, while Hexad C acts as an atrial activator for SMyHC III. Finally, the authors trace the evolutionary origins of cNRE and present evidence suggesting that it may originate from an endogenous viral element. Overall, this is a very impressive study that will contribute to a better understanding of the evolutionary pathway to achieve cardiac chamber-specific expression. However, some specific issues should be addressed before publication.

Major points:

To assess the sufficiency of cNRE for driving preferential atrial expression, the authors incorporated five tandem repeats of cNRE into the ventricular-specific vmhc promoter to examine its ability to bias ventricular-specific expression towards atrial activation. However, the results indicate that the vmhc promoter sequence appears to be somewhat shorter, as it did not exclusively drive vmhc expression in the ventricles. It is crucial to extend the vmhc promoter sequence to eliminate its atrial expression. The full 2.2 kb upstream of vmhc promoter may be necessary to ensure robust ventricular-specific expression (Zhang & Xu, 2009).

Minor points:

- 1) Figure 1F should be referenced within the manuscript.
- 2) The representation of dinucleotide substitutions (from '5'-GG-3' to '5'-TT-3') for Mut B and Mut C in the manuscript does not align with the labeling in Figure 2C and Figure 2I.

Author's answers to reviewer's questions, comments, and criticisms

We thank reviewer #2 for this kind appraisal of our contribution and for the comments relative to the vmhc promoter.

Major Points:

However, the results indicate that the vmhc promoter sequence appears to be somewhat shorter, as it did not exclusively drive vmhc expression in the ventricles. It is crucial to extend the vmhc promoter sequence to eliminate its atrial expression. The full 2.2 kb upstream of vmhc promoter may be necessary to ensure robust ventricular-specific expression (Zhang & Xu, 2009).

We would like to apologize to the reviewer for not having been clearer in our manuscript. Apparently, this lack of clarity caused some confusion. We have, in fact, used complete promoter of the zebrafish vmhc gene, which has been shown to drive strong ventricle-specific expression in zebrafish embryos at 48 hours post fertilization. Embryos at this stage expressing reporters with this promoter are also completely devoid of expression in the atrium. This has been reported, not by Zhang & Xu (2009), but by Jin et al. (2009). Our results are completely coherent with those reported by Jin et al. (2009). We have corrected this notion in the revised manuscript and have added the corresponding reference to the pertinent sections. Furthermore, to highlight the specificity of the construct, we included, in the revised manuscript, a new supplementary showing expression of vmhc::mCherry in stable zebrafish transgenics, highlighting the specificity of expression of this reporter line in the ventricle.

*Jin, D., Ni, T.T., Hou, J., Rellinger, E., Zhong, T.P. (2009). Promoter analysis of ventricular myosin heavy chain (vmhc) in zebrafish embryos. **Dev. Dyn.** 238, 1760–1767. doi: 10.1002/dvdy.22000*

*Zhang, R., Xu, X. (2009). Transient and transgenic analysis of the zebrafish ventricular myosin heavy chain (vmhc) promoter: an inhibitory mechanism of ventricle-specific gene expression. **Dev. Dyn.** 238, 1564–1573. doi: 10.1002/dvdy.21929*

Minor points:

1)Figure 1F should be referenced within the manuscript.

Thank you for pointing this out. Figure 1F is now referenced in the manuscript.

2)The representation of dinucleotide substitutions (from '5'-GG-3' to '5'-TT-3') for Mut B and Mut C in the manuscript does not align with the labeling in Figure 2C and Figure 2I.

Thank you for drawing our attention to this point. We have corrected the mistakes related to the dinucleotide substitutions throughout the manuscript.

Reviewer #3

1. What are the major claims of the paper?

This paper deals with cardiac gene regulatory networks. Specifically, the atrial-specific regulation of the Slow Myosin Heavy Chain III (SMYHC III) gene expression is examined. The authors claim to have discovered a third hexad sequence in the SMYHC III promoter that is responsible for atrial-specific gene regulation, and that this might have an evolutionary history dating back more than 60 million years ago, through a viral integration into an ancestral galliform bird host.

2. Are they novel and will they be of interest to others in the community and the wider field?

Yes - specifically for the those interested in developmental biology of how optimal cardiac function is regulated. In broader terms, the idea of ancient viral integration and changes in gene regulatory networks is likely to continue to be a common theme for many different systems. It is interesting to note that these repeats appear in the telomere regions.

3. If the conclusions are not original, it would be helpful if you could provide relevant references.

I think the conclusions are original - there's certainly lots of other papers about ancient viral integration, but I think the authors build a compelling story for this case.

4. Is the work convincing, and if not, what further evidence would be required to strengthen the conclusions?

Yes, it's convincing, and described clearly.

5. On a more subjective note, do you feel that the paper will influence thinking in the field?

Potentially it could. The authors ask a reasonable question - obviously the gene regulatory networks must somehow allow specific regulation of contractile proteins that are different for each chamber of the heart - how does this happen? And they come up with a testable hypothesis - the regulation is similar to what's already known (a purine-rich region, with two known Hexads) and for SMyHC III regulation, a 32 bp repeat with three purine-rich hexanucleotide repeats is responsible for chamber-specific gene expression regulation.

6. Please feel free to raise any further questions and concerns about the paper.

No further questions.

7. We would also be grateful if you could comment on the appropriateness and validity of any statistical analysis, as well the ability of a researcher to reproduce the work, given the level of detail provided.

I have no problems with the current use of statistics in this manuscript.

Author's answers to reviewer's questions, comments, and criticisms

We appreciate this reviewer's perception of the quality of our contribution. We would like to point out that this reviewer was the only one considering the totality of our work, not only a sub-section.

REVIEWERS' COMMENTS:

Reviewer #1 (Remarks to the Author):

This revised manuscript has been improved and the authors have attempted to address my major issues. Additional representative results are added to support the claims, binding of COUPTF-II to all three hexad elements is demonstrated to support the mechanism, and the authors significantly tempered claims of chamber "specificity" to more accurately describe the atrial "bias" particularly in the context of the fish transgenics.

Overall, the data and evolutionary comparisons are interesting and appropriate for publications.

Reviewer #2 (Remarks to the Author):

I have carefully reviewed the revised version of your manuscript titled "Unraveling the evolutionary origin of the complex Nuclear Receptor Element (cNRE), a cis-regulatory module required for preferential expression in the atrial chamber" and appreciate the effort you have put into addressing the previous comments and revising the content. Your feedback on the questions dispelled our concerns, and I have no further questions to ask.